# Metabolic and Cellular Compartments of Acetyl-CoA in the Healthy and Diseased Brain

**DOI:** 10.3390/ijms231710073

**Published:** 2022-09-03

**Authors:** Agnieszka Jankowska-Kulawy, Joanna Klimaszewska-Łata, Sylwia Gul-Hinc, Anna Ronowska, Andrzej Szutowicz

**Affiliations:** Department of Laboratory Medicine, Medical University of Gdansk, 80-210 Gdansk, Poland

**Keywords:** acetyl-CoA metabolism, neurodegenerative diseases, zinc dyshomeostasis, thiamine deficiency, aging

## Abstract

The human brain is characterised by the most diverse morphological, metabolic and functional structure among all body tissues. This is due to the existence of diverse neurons secreting various neurotransmitters and mutually modulating their own activity through thousands of pre- and postsynaptic interconnections in each neuron. Astroglial, microglial and oligodendroglial cells and neurons reciprocally regulate the metabolism of key energy substrates, thereby exerting several neuroprotective, neurotoxic and regulatory effects on neuronal viability and neurotransmitter functions. Maintenance of the pool of mitochondrial acetyl-CoA derived from glycolytic glucose metabolism is a key factor for neuronal survival. Thus, acetyl-CoA is regarded as a direct energy precursor through the TCA cycle and respiratory chain, thereby affecting brain cell viability. It is also used for hundreds of acetylation reactions, including N-acetyl aspartate synthesis in neuronal mitochondria, acetylcholine synthesis in cholinergic neurons, as well as divergent acetylations of several proteins, peptides, histones and low-molecular-weight species in all cellular compartments. Therefore, acetyl-CoA should be considered as the central point of metabolism maintaining equilibrium between anabolic and catabolic pathways in the brain. This review presents data supporting this thesis.

## 1. Introduction

Acetyl-CoA is a direct precursor substrate for the tricarboxylic acid (TCA) cycle coupled with energy production in the respiratory chain. In the brain, the largest fraction of acetyl-CoA is synthesised in mitochondria by the pyruvate dehydrogenase complex (PDHC) from pyruvate originating from glycolytic metabolism of glucose or from lactate oxidation by lactate dehydrogenase 1 (EC 1.1.1.1, LDH 1). The acetoacetate/β-hydroxybutyrate (AcAc/BHB), acetate, fatty acids and branched chain amino acids may provide smaller amounts of this metabolite in a concentration-dependent manner through β-ketothiolase (EC 2.3.1.16), acetyl-CoA synthase (ACCS, EC 6.2.1.1), beta-oxidation and branched ketoacids dehydrogenase pathways, respectively [1,2]. However, unlike peripheral tissues, the brain lacks significant metabolic flexibility, utilising plasma glucose as an obligatory, principal energy substrate. Non-glucose substrates cannot compensate for significant limitations in glucose supply apart from AcAc/BHB in starvation [1,2,3] (Figure 1). Nevertheless, irrespective of the precursor, acetyl-CoA is a merging endpoint metabolite, which enters the TCA cycle through a citrate synthase reaction. In resting conditions, the brain utilises 10 times more glucose and oxygen per weight than peripheral tissues; this is due to the continuous generation of action potentials in neurons in the range of 5–50 Hz. Moreover, restoration of neuronal membrane resting potential during each depolarisation/repolarisation cycle requires large amounts of energy. Hence, human brain neurons that make up 10% of all brain cells utilise 60–80% of glucose and oxygen. On the other hand, glial cells, which constitute 80–90% of human brain cells, produce 10% of the energy pool but export significant amounts of lactate to support neuronal energy metabolism. This is due to the prevalence of glycolysis over oxidative metabolism in astroglia and oligodendroglia [4,5,6]. Recent immunolabelling studies revealed that SOX9, an astrocyte-specific nuclear marker, is co-expressed with GLUT1 in astrocytes, constituting 10–20% of brain cells. However, they may be upregulated in several brain pathologies, including strokes, mini strokes and amyotrophic lateral sclerosis [7]. Moreover, physiological stimuli, such as feeding young mice with a high-fat diet for one month, increased astroglia size, branching and metabolism along with improving behaviour in animal behavioural tests [8].

Acetyl-CoA is also the substrate for hundreds of acetylation reactions catalysed by diverse acetyltransferases of different specificity, which can be found in each cellular sub-compartment. Products of acetylation include N-acetyl-aspartate and other acetylated amino acids and amines, such as acetylcholine, diverse acetylated proteins, carbohydrates and lipids, located in different cellular and subcellular compartments of the brain in concentrations varying from 10**^−^**^2^ to 10**^−^**^9^ mol/L. Acetyl-CoA is also the key substrate for the synthesis of structural and metabolic pools of fatty acids, cholesterol, phospholipids and other lipids. They are indispensable for neuronal myelinisation, cell growth, remodelling and maintenance of cellular membrane integrity [9,10]. Acetyl-CoA itself may also directly regulate rates of diverse acetylating pathways in a concentration-dependent manner. It also reciprocally affects its availability in various subcellular and cellular compartments of the brain [11,12] (Table 1). Thereby, acetyl-CoA should be considered as a branching point between the catabolic energy-providing TCA cycle and multiple acetyltransferases synthesising acetylated derivatives. It is not only the metabolic substrate, but it also plays the second messenger role and takes part in phenotypic and genotypic modifications connected with basic functions of the brain, as well as with its maturation and aging [13,14,15,16] (Figure 1). 

However, in the past century, reports on direct assessments of acetyl-CoA in the brain have been scarce due to complicated assay methods [17,18,19]. Recently, several publications described and discussed regulatory mechanisms of intercellular metabolic fluxes of glucose and other acetyl-CoA precursors, as well as their role in prevention against neurotoxic insults and exerting neuroprotective effects in humans and animal models of diverse brain pathologies [5,6,14]. The apparent diversity of acetyl-CoA metabolic pathways in the brain results from the existence of several distinct neuronal neurotransmitter systems, as well as astroglial, oligodendroglial and microglial cells and their specific functional status in a given region. The intracellular acetyl-CoA is unevenly distributed among mitochondrial, cytoplasmic, endoplasmic reticulum, nuclear and axonal sub-compartments [6,14]. This is due to the relative impermeability of intracellular membranes including specific carriers for this metabolite [14,15]. Alterations of the acetyl-CoA level in mitochondrial and cytoplasmic compartments have been shown to regulate viability and cholinergic neurotransmission of cholinergic septal SN56 cells. Cholinergic neurons of the basal forebrain are responsible for multiple cognitive functions, including working, episodic and spatial memory, learning, attention, behavioural phenomena and sensory information assuring selectivity and precision [6,13,14,20] (Figure 2). This review provides insight into quantitative and qualitative interactions between acetyl-CoA and metabolic pathways critical for neuronal and glial cell functions and their survival under physiologic and pathologic conditions. 

## 2. Glucose and Lactate—The Key Precursors of Brain Acetyl-CoA in Health and Disease

Glucose is a principal energy source that fuels 95% of ATP synthesis in the brain. Its transport from blood plasma through the blood–brain barrier into brain extracellular space is carried out by the specific, high-capacity, average-affinity 55 kD GLUT 1 transporter located on the outer side of vascular endothelium. Its constant affinity for glucose is equal to about 8 mmol/L. It achieves an inward transport rate dependent on systemic glucose concentration, diurnal physiologic variations of which may be in the range of 3.5–10 mmol/L. The density of 55 kD GLUT1 on the blood–brain barrier (BBB) capillaries is inversely regulated by glucose concentration in the plasma. Chronic hyperglycaemia results in an adaptative decrease in 55 kD GLUT1 density on the BBB, which reduces glucose transport into the brain [21,22,23]. In turn, in chronic hypoglycaemia, hypoxia and several acquired and congenital metabolic diseases, the density of 55 kD GLUT1 increases, adjusting in part the rate of inward transport of glucose to these conditions [24,25]. Upregulation of GLUT1 is mediated by the phosphatidylinositol-3-kinase, AMP-activated protein kinase, cAMP response element–binding protein and hypoxia inducible factor pathways. On the other hand, the mitogen-activated protein kinase pathway downregulates GLUT 1 and 3 expression [25]. Such mechanisms tend to reduce concentration-dependent fluctuations in the rate of glucose transport into the brain. In fact, some of those patients display good tolerance to hypoglycaemia (Figure 1). 

Neurons take up glucose from extracellular brain space mainly by the high-affinity GLUT3 transporter with Km about 2.8 mM, which may secure its appropriate supply in moderate hypoglycaemia [21,23]. Pyruvate derived from glycolysis or from oxidation of exogenous or endogenous lactate may serve as a direct precursor of neuronal acetyl-CoA. Lactate is released from astroglial or oligodendroglial cells by MCT4 transporters with low affinity (Km 22–28 mM) for lactate and transported into neuronal cell body axons and presynaptic terminals, respectively. Neurons express high-affinity MCT2 transporter with Km for lactate and pyruvate equal to 0.5–0.75 and 0.08 mmol/L, respectively; this promotes the direction of lactate/pyruvate transport toward neuronal cells [23,26,27]. Such direction of metabolic flow is also facilitated by the fact that LDH-1 isoform, which is primarily expressed in neurons, oxidises lactate to pyruvate, whereas the LDH-5 isoform expressed in astrocytes promotes the reduction of pyruvate to lactate. Despite their distinct distribution, LDH-1 and LDH-5 do not directly modulate the lactate flow between neurons and astrocytes [6]. Nevertheless, LDH5 would facilitate lactate accumulation and its subsequent release from astroglia, whereas LDH1 would support the metabolism of lactate directly towards pyruvate and its oxidative decarboxylation (Figure 1).

However, in vivo lactate cannot replace glucose entirely as an energy precursor either in physiological or in pathological conditions. Lactate at high-physiologic 1 mmol/L concentration can substitute only about 10% of glucose. However, during lactic acidosis evoked by intensive exercises, hypoxia, diabetes, chronic obstructive lung disease, disseminated cancer or inherited metabolic diseases, blood lactate may reach 10 mmol/L and higher concentrations and replace up to 25% glucose in acetyl-CoA and energy production [28]. On the other hand, cultured primary neurons and neuronal stem cells survived in glucose-free medium in the presence of lactate as the only energy substrate [29]. Moreover, SN56 cholinergic neuronal cells utilising pyruvate with malate as sole energy substrates retained stable levels of acetyl-CoA and structural integrity [14].

Various types of neurons and glial cells may respond differentially to the same neurotoxic conditions. For instance, in chronically hypoxic BV2 microglial cells, the acetyl-CoA level was three times higher than in normoxic ones [30]. This apparent inconsistency was explained by a hypoxia-induced increase in mRNA and protein levels of hexokinase 2 (HK2, EC 2.7.1.1), yielding activation of the glycolytic pathway. This led to the rise of pyruvate synthesis, which through the PDHC provided more acetyl-CoA for acetylations of nuclear histones 3 and 4 [31,32,33]. Acetylated histones induced the inflammatory phenotype in BV2 by elevating proinflammatory gene expression followed by augmentation of CD11 and IL-1β levels. Inhibition of HK2 or PDHC activities by their specific inhibitors, lonidamine or 3-bromopyruvate, decreased acetyl-CoA levels and alleviated proinflammatory responses of cultured BV-2 microglial cells. Similar effects were observed after knockdown of HK2 with specific siRNA, but not after suppression of HK1 and HK3. This proves that only the increase in HK2 expression was responsible for the over-activation of BV2 cells. In addition, in in vivo experiments, suppressed activity of microglial HK2 by pretreatment with lonidamine reduced the size of ischemic injury in the rat brain in the middle carotid artery occlusion model (MCAO) [29]. MCAO followed by 40 min of hypoxia in 9-day-old neonatal mice after 24 h brought about a decrease in PDHC activity but no changes in the acetyl-CoA level in crude mitochondrial fraction. Animals pretreated with dichloroacetate (DCA), a pyruvate dehydrogenase kinase (PDK, EC 2.7.11.2) inhibitor, showed increased activity of the PDHC and levels of acetyl-CoA above the control values, as well as reduced cell apoptosis [34] (Table 1). MCAO-induced failure of energy metabolism in the brain is claimed to be overcome with application of Shengui Sanhseng San extract [35] (Table 1). This preparation has been used for over 300 years by Traditional Chinese medicine to treat stroke. It consists of a mixture of rhizome and roots of three plants. Its application for 7 days to MCAO animals increased, in a dose-dependent manner, the density of GLUT1 and 3 and the levels of pyruvate, citrate, acetyl-CoA and ATP and decreased phosphorylation of E1 subunit PDHC in the post-ischemic region of the brain [35]. A pleiotropic, neuroprotective effect of resveratrol preconditioning was also reported. It increased ischemic tolerance in in vivo and in vitro models [36,37]. Neuronal–astrocytic co-cultures obtained from 14 d resveratrol-treated mice displayed increased expression of mitochondrial pyruvate carriers and citrate synthase levels, yielding an increase in citrate and ATP synthesis and a delay of excitotoxic injury during oxygen–glucose deprivation [36]. Such conditions increased the passage of citrate into the nucleus, which augmented its conversion to acetyl-CoA by nuclear ACLY and activated acetylation of histones H3K9ac and H4K16ac by HAT [36]. It should be emphasised that, although the presented data strongly suggested the involvement of acetyl-CoA in resveratrol-mediated neuroprotection, no direct findings concerning its mechanisms have been presented [36,37] (Figure 1). 

On the other hand, modest inhibition of PDHC activity in cultured N9 microglial, C6 astroglial and primary glial cells by 0.15 mmol/L Zn or NO excess brought about no significant decreases in their acetyl-CoA/ATP levels or their viability [11,38,39]. On the contrary, under similar culture conditions, Zn caused over 50% suppression of PDHC activity and acetyl-CoA content in differentiated cholinergic SN56 neuronal cells (Table 1). These alterations were accompanied by a significant loss in rates of ACh synthesis/release and neuronal viability [11,39,40]. The presented data indicate that adaptive overexpression of glycolytic pathway in glial cells, after hypoxic insult, may overcome inhibition of acetyl-CoA synthesis caused by suppression of PDHC activity due to provision of additional amounts of pyruvate [30]. Another cause of relative resistance of neuroglia to neurotoxic conditions may be their lower energy demands for maintenance of plasma membrane potential and non-utilisation of acetyl-CoA for neurotransmitter or NAA synthesis [11,14,30]. On the other hand, energy requirements for neurons are much higher than those for glial cells. Therefore, inhibition of PDHC activity exerted deeper suppressive effects on acetyl-CoA content and viability in neuronal than in glial cells [38,39] (Table 1). 

There are data showing that disturbances in brain Zn homeostasis may be the primary cause of neurodegeneration. Open-head traumatic injury of the brain cortex caused immediate hyperglycaemia lasting up to 3 h, followed by severe hypoglycaemia in both male and female rats. In the ninth post-trauma hour, loss of respiratory control was observed in isolated brain mitochondria, along with suppression in acetyl-CoA and ATP levels in the peri-contusional ipsilateral cortex of male rats [41] (Table 1). These changes could be evoked by excitotoxic effects of Zn released in excess from impaired presynaptic terminals [42,43]; then, Zn was taken up by postsynaptic neurons causing the inhibition of the PDHC and several enzymes of the TCA cycle, ultimately leading to their death [39,44,45]. These pathologic alterations could be corrected by early i.v. infusion of lactate or a delayed one of BHB that maintained their stable blood concentrations of 1.2 and 2.0 mmol/L, respectively. In female rats, contusion brought about inhibited respiration rates but no changes in acetyl-CoA and ATP levels. Early infusion of 2 mol/L BHB increased acetyl-CoA and ATP over control levels, whereas 100 mmol/L lactate was without effect. On the contrary, early infusion of BHB appeared to be harmful, resulting in severe decreases in acetyl-CoA and ATP levels in peri-contusional tissue. This indicates that, in traumatic brain injury, therapeutic i.v. applications of BHB or lactate as complementary to glucose direct precursors of acetyl-CoA should consider the post-trauma time and sex of the patient to avoid negative side effects of such treatment [41]. However, there is no rational explanation for post-trauma time and sex-linked differences in the beneficial or harmful effects of exogenous lactate or HB on acetyl-CoA-mediated post-traumatic recuperation of the brain. Nevertheless, irrespective of the particular pathomechanism, the beneficial effects of lactate or BHB were accompanied by an increase in the whole tissue acetyl-CoA level, indicating a key role of this intermediate in brain healing [40]. Such a thesis is supported by the results of clinical trials that revealed beneficial effects of infusion hyperosmic sodium lactate, being a direct acetyl-CoA precursor. It bypasses glycolysis as the ATP consuming pathway and alleviates reperfusion injury in patients after focal cerebral ischemia [46]. The positive influences of hypertonic lactate infusion in brains of TBI patients or those with acute cardiac failure may be extended beyond its role as an alternative energy precursor, as it is also an anti-oedematous agent, scavenger of free radicals, Zn/Ca chelator and suppressor of reperfusion-evoked glutamate/Zn/NO excitotoxicity [46,47]. For obvious reasons, the effects of lactate/HB on the acetyl-CoA status in injured brains were not investigated in humans. 

Cardiac arrest is a prevalent cause of death worldwide. Cardiopulmonary resuscitation has improved survival, but many patients die soon after due to anoxic brain injury and cardiac instability [48]. In rats subjected to 6 min anoxia and then resuscitated, intraperitoneal injection of dichloroacetate (DCA) caused a two-fold increase in the survival rate and alleviated neurological deficits during the 72 h post-resuscitation period. DCA decreased levels of proinflammatory IL-1β and TNF-α and blood lactate concentration, along with partial restoring decreased PDHC activity and ATP and pyruvate levels in the hippocampus and brain cortex [48]. The inhibition of the PDHC may be caused by an increase in intraneuronal Ca levels due to its excessive influx through voltage-gated calcium channels and NMDA receptors in depolarised plasma membranes. Ca excess activated PDK, yielding inhibitory phosphorylation of the E1 PDHC subunit [49,50]. These data suggest that the beneficial effects of DCA resulted from the inhibition of PDK, yielding the dephosphorylation of the E1 subunit and an increase in PDHC activity, followed by increases in pyruvate oxidation and acetyl-CoA and ATP levels [48,49,50]. Moreover, in mice, therapeutic hypothermia or DCA application inhibited the PDK and reactivated the PDHC, improving outcome after cardiac arrest [51]. Cardiac arrest in mice lasting 8 min followed by cardiopulmonary resuscitation resulted in a rapid and deepening-with-time decrease in thiamine diphosphate (TDP) and ATP levels in the brain cortex, resulting in a high rate of animal mortality. This was due to phosphorylation-induced inhibition of PDHC activity, yielding a deficit of acetyl-CoA in neurons [46,52]. Daily i.v. supplementation of thiamine increased TDP and ATP levels in brains, decreasing the rate of animal mortality [52]. Moreover, in humans, cardiac arrest caused a marked acute decrease in PDHC activity in mononuclear blood cells, which was partially restored within 72 h after resuscitation. In addition, 30 min of severe hypoglycaemia in rats after 7 days brought about a several-fold increase in PDK2 and inhibition of PDHC activities in the brain, causing extensive neuronal death, along with activation of astroglia and microglia [53]. These pathologic alterations were alleviated in part by injections of DCA. In summary, these data prove that the PDHC is a common target for several neurodegenerative signals. Therefore, maintenance of the activity of this complex should be considered as the potential goal for neuroprotective interventions. Moreover, these data suggest that the inhibition of PDK or an increase in thiamine supplementation may benefit neuroprotective treatment of patients after cardiac arrest [51,52].

Other in vivo experiments revealed mechanisms that may partially compensate for shortages of acetyl-CoA evoked by deficits of the PDHC. In brain-specific heterozygotic Pdha1 knockdown mice, PDHC activity was reduced by 68%, but the acetyl-CoA level was not significantly decreased against wild-type controls [54] (Table 1). Such a discrepancy may be explained by the compensatory activation of the ACSS1 pathway, which increased acetate incorporation into acetyl-CoA in animal brains. Systemic administration of acetate to PDHC-deficient mice stimulated metabolic flux through the TCA cycle and normalised glutamatergic neurotransmission, yielding suppression of gamma oscillations and epileptiform discharges [54,55]. Beneficial effects of dietary supplementation with glyceryl triacetate—as a source of acetate—were also observed in an experimental autoimmune encephalomyelitis (EAE) mouse model of multiple sclerosis. Such treatment prevented the loss of ethanolamine, phosphatidyl choline and cholesterol in the myelin of EAE mice compared to EAE controls treated with water [56]. These data suggest that exogenous acetate or its donors may be used as a complementary precursor of acetyl-CoA, bypassing the PDHC step, to increase lipid deposition in oligodendrocytes and neurons impaired by demyelinating diseases [56] (see chapter 6 for details). Such a conclusion is supported by the finding that, in mice with specifically deleted Pdha1 gene, Schwann’s cells and brain oligodendrocytes retain capacity for myelinisation. This may be caused by the existence of compensatory overexpression of the ACSS1 pathway providing acetyl-CoA, which bypasses the PDHC step in oligodendrocytes [54,57]. On the other hand, animals with Pdha1 deletion in all brain cells displayed reduced fibre density and signs of axonal degeneration. This suggests that acetyl units in PDHC-deficient oligodendrocytes are provided by adjacent PDHC-competent astroglial and neuronal cells [54,57].

TBI suppressed PDHC expression in the peri-contusional area, and the rate of this decline depended on the potency of the impact. Mild or severe TBI brought about differential effects on genes expression, protein levels and the activities of several enzymes linked with energy production [58] (Table 1). Mild TBI within 5 days post-trauma caused increases in gene expression of catalytic E1, E2 and E3 PDHC subunits, along with decreases in PDK and gradual, delayed increases in PDP expression. Such a profile of PDHC subunits would be compatible with a stable level of acetyl-CoA in a mildly insulted brain. On the other hand, severe TBI did not affect expression of PDHC catalytic subunits but strongly suppressed PDP (down to 5% of controls) and elevated PDK gene expression. Such a pattern promoted inhibition of the PDHC E1 subunit due to PDK-dependent inhibitory phosphorylation [50]. Accordingly, marked decreases in acetyl-CoA/CoA-SH levels were found in cases of severe TBI [58]. Hence, post-traumatic differential changes in brain acetyl-CoA levels in severe vs. mild TBI result from variations in phosphorylation/dephosphorylation levels yielding the inhibition or activation of the PDHC, respectively. A higher level of acetyl-CoA may be a predictive marker for positive outcome following TBI [58].

Inhibition of energy metabolism is an early sign of mitochondrial dysfunction in AD and other neurodegenerative diseases; it is caused by decreases of activities or expression of the PDHC, KDHC and some other mitochondrial enzymes of the TCA cycle evoked by accumulating Aβ and hyperphosphorylated tau [59,60]. Aβ1-42 in sub-micromolar concentrations was found to inhibit activities of the PDHC and KDHC in vitro in synaptosomes isolated from the brain of WT rats or in synaptosomes from 2756 Tg AD mice and in clonal neuronal cells [14,61,62]. However, contradictory results were presented by Gandbhir and Sundaram [63]. They showed that a relatively high 0.004 mM concentration of pre-aggregated Aβ1-42 markedly increased levels of PDHC and KDHC proteins, causing tau hyperphosphorylation and impairment of the SH-SY5Y cholinergic cell line. These effects were partially reversed by AmyTrap, a homo-tetrameric peptide synthesised from D-amino acids, which could remove Aβ1-42 from neuronal cells [63]. However, it remains unknown as to how Aβ-induced increases of PDHC and KDHC expression resulting in elevations of acetyl-CoA and ATP levels could exert neurotoxic effects. These data contradict the well-documented view that high levels of ATP and acetyl-CoA are markers of the wellbeing of neurons and other cells [13,14,15,53,64]. For an explanation of this discrepancy, determinations of in situ enzyme activities, as well as estimations of ATP and acetyl-CoA levels in different culture conditions, are necessary. Such information is indispensable in light of the claimed therapeutic potential of the AmyTrap compound [63].

## 3. Origin and Metabolic Role of Axonal Acetyl-CoA

Axons play an important role in the maintenance of neuronal integrity. Axonal transport is a process that delivers organelles, proteins and low-molecular-weight species, occluded in axonal vesicles, along microtubular tracks to distant neuronal axonal compartments; they also convey retrograde transport. Neurotubules consist of globular α and β tubulin heterodimer subunits forming helical chains extending along entire axons. They are stabilised by tau protein and α-tubulin lysine residues acetylated by specific α-tubulin acetyl transferase (EC 2.3.1.108, αTAT) [65]. Microtubular transport also takes place in growing dendrites and axons during brain development. In the adult brain, axonal transport is indispensable for survival, integrity and proper neuronal function. Microtubules are involved in the bidirectional transport of proteins and trophic factors, signalling peptides and organelles, including mitochondria, between neuronal perikaryon and nerve terminals [64,66] (Figure 3). The biogenesis of mitochondria takes place in neuronal perikaryons, as the majority of mitochondrial proteins are encoded by nuclear DNA. They have to be transported through the axonal microtubular system to nerve terminals, which produce and utilise about 55% whole neuronal ATP to maintain membrane potential, synaptic vesicle recycling and stabilisation of the neurotransmitter pool [67,68]. Because of their length, varying from a few micrometres to several centimetres, axons need on-site energy synthesis by local axonal mitochondria [67]. Axons are insulated from extracellular fluid by several layers of myelin sheets formed by oligodendroglia that limit their access to extracellular glucose and lactate. Therefore, lactate is supplied by surrounding oligodendroglial processes through their MCT1 to peri-axonal space, from where it is taken up by axonal MCT2 and used for acetyl-CoA and ATP synthesis in axonal mitochondria [69,70]. Fractions of lactate and glucose are also provided by astrocytes that contact axons at nodes of Ranvier’s. They shuttle lactate and other metabolic substrates through astrocytic MCT1, MCT4 and neuronal MCT2 (Figure 3). Thus, alterations in the provision of energy substrates to axonal mitochondria are a key factor in maintenance and triggering neurodegeneration and regeneration [68]. 

The impairment of oligodendroglia in multiple sclerosis by autoantibodies against myelin basic protein caused early loss of MCT1 and failure of metabolic support to axons, which also displayed downregulation of MCT2 and GLUT3 [71,72]. A deficit of lactate supply abrogates energy production, axonal transport and depolarising of signal conductance. This may cause early axonal deterioration that may precede demyelination. On the other hand, expression of MCT1 in astrocytes seems to be upregulated due to inflammatory activation [71,72]. One can predict that such disturbances should decrease the acetyl-CoA level in this neuronal compartment although no direct data are available. Nevertheless, they demonstrate the importance of oligodendroglial and astroglial metabolic support with acetyl-CoA precursors to support axono-neuronal integrity [73] (Figure 3). Acetylation of alpha-tubulin units plays an important role in neurotubular stabilisation and the stimulation rate of axonal growth. It is catalysed by axon-specific αTAT 1, which binds directly to neurotubules, facilitating their acetylation. Transfection of cultured DRG neurons or clonal T293 cells with the αTAT1 gene caused a several-fold increase in tubulin acetylation, yielding elongation of axons. Co-transfection with the histone deacetylase 5 (EC 3.5.1.98) gene decreased αTAT1 and evoked α-tubulin acetylation but did not affect axonal growth [74]. Moreover, overexpression of these proteins in rat DRG had no effect on sciatic nerve regeneration in vivo. Studies of cultured primary cortical neurons from embryonic mice revealed that overexpressed αTAT1 both acetylated tubulin and overcame axon degeneration caused by chondroitin sulphate proteoglycans that reduced α-tubulin acetylation. These data define αTAT1 as a potential target to alleviate brain injury, which increases levels of toxic substrates abrogating axonal regeneration [75].

Both anterograde and retrograde axonal transport are executed by motor proteins kinesin and dynein coupled with an elongator complex, which supports α-tubulin acetylation catalysed by αTAT1 located in axonal vesicles (Figure 3). Acetyl-CoA for acetylation is provided by ATP-citrate lyase (ACLY, EC 2.3.3.8), which is also enriched in this sub-compartment [76]. Loss of elongator activity in ELP3 KO neurons reduced ACL activity and the level of tubulin acetylation, yielding an inhibition of axonal transport in isolated cortical neurons. Moreover, inhibition of ACLY by (-)hydroxycitrate suppressed α-tubulin acetylation due to a shortage of acetyl-CoA in axonal cytoplasm [76]. These data indicate that the elongator ACLY-αTAT1 complex plays a key role in the regulation of neurotubular acetyl-CoA-dependent transport in neuronal axons. Inherited or acquired defects of the elongator complex are a hallmark of diverse neurodegenerative diseases [77]. Such disturbances were found in fibroblasts from familiar dysautonomia patients, which is brought about by autosomal, recessive mutation of the elongator subunit gene ELP1 [76] (Table 1).

## 4. Glucose and Pyruvate-Derived Acetyl-CoA Metabolism in Cholinergic Neurons

ACh is phylogenetically the oldest neurotransmitter and is synthesised by choline acetyltransferase (ChAT, EC 2.3.1.6) expressed exclusively in the cytoplasm of cholinergic neurons. Along with the high-affinity choline uptake system, vesicular ACh transporter, M2 autoreceptor and ACh, ChAT constitutes the core of the integrated system responsible for synthesis, intra-vesicular accumulation and quantal release of this neurotransmitter. They are biomarkers of the structural integrity and functional competence of cholinergic neurons [15,78].

Pyruvate is the key precursor of acetyl-CoA in neuronal mitochondria and is utilised mainly for intramitochondrial energy production and N-acetyl-aspartate synthesis. About 5% of acetyl-CoA generated in the mitochondria has to be transported through the mitochondrial membrane to the cytoplasm to support diverse acetylation pathways in all neurons, and additionally, for neurotransmitter–ACh synthesis taking place exclusively in cholinergic neurons, which constitute a small fraction of brain neurons [15,78].

In non-depolarised cells, the inner mitochondrial membrane is impermeable for acetyl-CoA and other acyl-CoA derivatives. Therefore, acetyl units have to be transported from the mitochondria to the cytoplasm indirectly: (i) after conversion to citrate via citrate synthase (EC 2.3.3.1) through the SLC25A1/CIC malate-citrate antiporter on the inner mitochondrial membrane and cytoplasmic ACLY; or (ii) through the mitochondrial acetyl-CoA hydrolase (EC 3.1.2.1), monocarboxylate transporter 1 and cytoplasmic ACSS1 and (iii) mitochondrial carnitine acetyltransferase 1, 2 (EC 2.3.1.7) pathways, respectively [13]. In depolarised nerve terminals and cultured cholinergic neuronal cells, acetyl-CoA may be transported to the cytoplasm directly through transient, depolarisation-Ca**^2+^** activated permeability transition pores (PTPs) [19,79].

### 4.1. Acetyl-CoA Compartments in Cholinergic and Noncholinergic Neurons

The level of intramitochondrial acetyl-CoA directly regulates the rates of its indirect and direct efflux from the mitochondria, thereby determining the level of acetyl-CoA in the cytoplasm and its utilisation in diverse cytoplasmic and nuclear acetylations [14]. The contribution of a particular pathway of acetyl-CoA transport to the cytoplasm may vary depending on the cell type, its functional status and extracellular levels of energy substrates (Figure 2).

Physiological aging is, among others, associated with the dysfunction of rat brain mitochondria due to decreasing PDHC activity, caused by increased phosphorylation of its E1 (pyruvate dehydrogenase EC 1.2.4.1) subunit by PDHK, whose expression and activity increase with age [80]. This causes a reduction in acetyl-CoA synthesis, which in turn diminishes ATP generation in the TCA/oxidative phosphorylation cycle in mitochondria, yielding a physiological age-related lowering of brain plasticity and an increased propensity to neurodegeneration [80,81]. One should stress that the activity of the PDHC and expression of E1 in WT mice brains were not changed at up to 12 months of age. On the other hand, PDHC activity and E1 level were significantly decreased in the brains of 3xTg AD mice of the same age, indicating the existence of structural shortages of acetyl-CoA [82]. Cholinergic neurons are particularly prone to degeneration due to their demand for additional acetyl-CoA groups for ACh and N-acetyl-L-aspartate (NAA) synthesis [14,15,83]. Therefore, a higher expression of the ChAT pathway in differentiated cholinergic neurons made them more susceptible to several age-related conditions, such as decreased synthesis of glutathione and other reducing compounds, episodic or chronic hypoxia, excessive glutamate and Zn release from glutamatergic nerve terminals and free radical excess [82,84]. Losses of cholinergic neuronal bodies in basal nuclei and their cortical projections are characteristic features of inherited and sporadic AD, TD, brain hypo-perfusion encephalopathy and other neurodegenerative conditions common in elderly people [85].

Culture of cAMP/RA differentiated SN56 septal cholinergic neuronal cells revealed that early and late excitotoxicity-induced compounds, such as NO, NOO^−^ and Zn^2+^, along with Aβ1-42 or exogenous Aβ25-35, inhibited PDHC activity, yielding decreases in intra-mitochondrial acetyl-CoA followed by a drop of its concentration in the cytoplasm and suppression of ACh synthesis, accumulation and quantal release [11,14,38,84]. In addition, nerve terminals isolated from the brains of aged, 15-month-old, Tg 2756 AD mice with cognitive deficits and an Aβ1-42 load of about 0.6 μM displayed a 46% decrease in pyruvate oxidation and proportional declines of mitochondrial and synaptoplasmic acetyl-CoA levels [62] (Table 1, Figure 2). Accordingly, ACh content as well as rates of its synthesis and quantal release were also depressed. Conversely, in whole brain mitochondria, pyruvate oxidation increased at an unchanged acetyl-CoA level, probably reflecting an inflammatory reaction of neuroglia to accumulated Aβ [29,62] (Table 1). They indicate that pathological regulations of PDHC activity may be different in individual neuronal and glial compartments of the brain [78]. However, in Tg 2576 AD mice, PDHC and ChAT activities in lysed synaptosomes were the same as in non-Tg controls when assayed at optimal substrate concentrations. This indicates that, in these mice, accumulated Aβ1-42 evoked purely functional cholinergic deficits without compromising the structure of cholinergic neurons in the brains. They were presumably caused by Aβ-induced inhibitory phosphorylation of the E1 subunit of PDHC, yielding decreases in the level of mitochondrial acetyl-CoA available for transport to the synaptoplasm [62]. These data are comparable with other studies on Tg 2576 mice, which have shown no decrease in ChAT activity and a marked decrease in ACh release rate against WT litter mates [86]. On the other hand, a significant 30–50% lowering of ChAT/AChE-positive neurons was observed in the basal forebrain, hippocampus and cortex of transgenic homozygous 3–9-month-old female mice expressing a truncated tau fragment [87]. A profound decrease in ChAT expression (80%) also took place in the basal forebrain but not in the hippocampus of APPswe/PSEN1dE 5-month-old demented mice. In the latter region, age-dependent suppression of cognition-related proteins—but not ChAT—were claimed to be responsible for the impairment of cognitive functions [88]. On the other hand, the brains of 3xTg-AD female mice displayed age-dependent decreases of both activity and levels of all PDHC subunits and mitochondrial oxidation in whole brain homogenates, which preceded the onset of Alzheimer’s Aβ/tau pathology [82,89]. They support the general conclusion that mitochondrial dysfunction, which generates acetyl-CoA and energy deficits, is an early signal triggering cognitive losses before the appearance of AD proteinopathy [60]. 

However, Tg mice carrying other APPswe/PS1dE9 or double-mutated tau displayed a marked decrease in several cholinergic biomarkers, including ChAT activity, ACh synthesis, levels of M2 autoreceptors and vesicular ACh transporter, as well as a reduction in the density of ChAT immunopositive neurons in the cortex and hippocampus accompanied by cognitive deficits [87,90]. These negative alterations were prevented by transplantation of cholinergic human stem cells. The presented data indicate that various combinations of human APP/PSEN/tau mutated genes determine the variability of the onset time and severity of cognitive deficits, as well as the sequence and site of appearance losses in energy synthesis (acetyl-CoA, ATP), cholinergic phenotypes and Aβ accumulation in various Tg models of AD [62,82,87,88]. Similar variability in the course of impairment of different steps of glucose to ACh pathways are likely to occur in individual cases of inherited AD patients. This thesis is supported by clinical MRI/PET studies with **^18^**F labelled Aβ ligands in the brains of patients with inherited AD, which displayed Aβ accumulation accompanied by decreases in **^18^**F-deoxyglucose uptake [84,91]. However, accumulation of Aβ in the brains of sporadic AD patients does not seem to be conclusive, because about 30% of cognitively competent old people also demonstrated an accumulation of Aβ. Nevertheless, co-accumulation of Aβ and phosphorylated tau protein preferentially impaired the structure and function of intrasynaptosomal mitochondria along with a fall in ^18^F-deoxyglucose uptake in brain regions affected by the AD [92,93]. These studies remain in accord with the existence of tight interdependencies between rates of acetyl-CoA synthesis by the PDHC in synaptosomal mitochondria, level of acetyl-CoA in synaptoplasm and rates of ACh synthesis/release linked with cognitive dysfunctions in aged Tg 2756-AD mice [14,62,94]. In fact, in AD patients, the failure of **^18^**F-deoxyglucose uptake in different cortical and subcortical structures, as seen in MRI/PET studies, correlated well with deficits in specific cognitive functions shortly before death, as well as with losses of cholinergic markers detected in post-mortem studies [95,96]. These findings indicate that the PDHC is a rate-limiting step regulating acetyl-CoA provision by pyruvate derived from the glycolytic pathway, thereby affecting ACh synthesis and rates of its release [13,14] (Figure 2). Decreases in PDHC activity or pyruvate uptake are frequent findings in cellular and animal models of encephalopathies and human AD brains as well [14,59,62,82]. As such, equilibrium between rates of glycolysis-linked synthesis of acetyl-CoA and its multidirectional utilisation appears to be a key factor for maintenance of the functional and structural integrity of neurons and glial cells [30] (Figure 1).

This thesis is supported by the fact that the maturation of neuronal cells resulted in significant changes in acetyl-CoA distribution. Thus, differentiation of cholinergic SN56 cells induced by cAMP/RA resulted in a deep decrease in acetyl-CoA levels in mitochondria and a marked increase in the cytoplasm and no significant changes in PDHC activity. Simultaneously, ChAT activity and ACh synthesis increased by 110 and 290%, respectively [97,98] (Table 1). Such a significant redistribution of intraneuronal acetyl-CoA could be due to the facilitation of its transfer from the mitochondrial to cytoplasmic compartment to meet increased demand for acetyl units for stimulated ACh synthesis in mature cholinergic neurons. In turn, a non-proportionally high increase in ACh synthesis against ChAT activity in differentiated neurons could result from a synergistic interaction of independent increases in ChAT activity and acetyl-CoA concentration in the cytoplasm, respectively [15,98]. Simultaneously, a relative shortage of acetyl-CoA in the mitochondria of differentiated cholinergic cells made them highly susceptible to different neurodegenerative signals that inhibited acetyl-CoA synthesis by the PDHC. Under such conditions, cell mortality displayed an inverse correlation with mitochondrial acetyl-CoA concentration [14]. In addition, post-mortem studies of AD brains revealed the existence of widespread, severe 50–70% deficits of pantothenate, which is an obligate precursor of CoA-SH [99]. 

These findings support the claim that several mechanisms generating acetyl-CoA deficits may take part in the preferential loss of cholinergic neurotransmission in the brains of AD patients and AD-Tg animals [81]. In fact, competition for mitochondrial acetyl-CoA was found to exist between the TCA cycle and the ACh synthesising pathway in SN56 neuronal cells with a high expression of cholinergic phenotype. The increase in cholinergic phenotype and ACh synthesis following treatment with cAMP-retinoic acid, NGF or CATgene-transfection of nondifferentiated SN56 cholinergic neurons was accompanied by a decrease in their acetyl-CoA without affecting viability in basal conditions [100] (Table 1). However, SN56 cholinergic neurons with a high expression of cholinergic phenotype appeared to be more susceptible than nondifferentiated ones or glial cells to several neurotoxic signals that inhibited the PDHC, resulting in the suppression of acetyl-CoA synthesis in mitochondria. Such alterations took place in cholinergic neurons or brain nerve terminals upon exposure to several common neurotoxic signals, such as Aβ, Zn, NO-excess, Ca overload, thiamine deficiency, aluminium exposure and hypoxia [11,14,40,45,98,100]. In all these conditions, decreases in pyruvate utilisation correlated with respective depressions of acetyl-CoA levels, ACh synthesis and cell viability [14]. This may explain the preferential loss of cholinergic neurons and respective cognitive functions in several neurodegenerative conditions, including Alzheimer’s, and it would justify the development of neuroprotective therapies targeting cholinergic neurons [101].

### 4.2. Zinc and Cholinergic Acetyl-CoA Metabolism

Zinc is an essential trace metal that plays a significant role in several processes, such as cell growth and differentiation, stabilisation of nucleic acid and protein structure, cellular signalling and neurotransmission [42,43,84,102]. Conversely, Zn dyshomeostasis may contribute to pathomechanisms of numerous brain diseases [103]. Average Zn concentration in the brain was estimated to be about 0.15 mmol/L. However, intracellular concentration of weakly bound and free Zn^2+^ is in the nanomolar to sub-micromolar range. This is due to the fact that 70–90% of the cation is stably bound with about 2000 proteins regulating the expression of 900 genes [102]. However, in synaptic vesicles of glutaminergic terminals, Zn loosely chelates with glutamate, reaching 1.0 mmol/L concentration in this compartment. During the depolarisation of glutaminergic terminals, Zn is co-released with glutamate to the synaptic cleft, where its concentration may transiently increase up to 0.30 mmol/L [43,84]. Protein concentration in interstitial fluid is estimated to be in range of 0.3–0.6 g/L. On the other hand, 1.0 g/L serum proteins bind about 0.015–0.02 mmol/L of Zn [40]. Therefore, a large, over 90% fraction of Zn released to the synaptic cleft occurs in the form of free cation. Unbound Zn^2+^ is cleared from the synaptic cleft by postsynaptic neurons and peri-synaptic neuroglial cells [104]. In pathologic conditions, prolonged depolarisation of glutaminergic terminals brings about the release of excessive amounts of glutamate and Zn to the synaptic cleft. This causes excitotoxic depolarisation of postsynaptic neurons through NMDA/AMPA receptors, which activate the accumulation of Zn and Ca excess through voltage-gated Ca-channels and specific Zn inward transporters [105]. 

Studies on DC SN56 cholinergic cells have shown that depolarisation caused a several-fold elevation of Zn accumulation. Hence, at 0.01 and 0.05 mmol/L concentrations in culture medium, Zn^2+^ slowly accumulated in non-depolarised neuronal cells, reaching intracellular levels of 0.28 and 1.35 mmol/L after 24 h incubation, respectively. On the other hand, depolarisation with 30 mmol/L K^+^ resulted in rapid Zn accumulation that reached a 4.0 mmol/L level within 30 min, causing instant neuronal death [105]. This resulted from the inhibition of PDHC and KDHC activities, yielding a decrease in acetyl-CoA concentrations, and its slower metabolic flow through the TCA cycle, yielding a decrease in ATP synthesis in neuronal mitochondria [39,105] (Table 1). The Zn-evoked deficit of acetyl-CoA in the mitochondria limited its efflux to the cytoplasm. As a result, lowered levels of acetyl-CoA in both compartments caused an increase in the neuronal death rate and an inhibition of cholinergic transmission, respectively. Zn directly inhibited PDHC and KDHC activities through competitive removal of lipoamide from their binding sites of E2 (dihydrolipoyl transacetylase, EC 2.3.1.12 and dihydrolipoyllysine-residue succinyltransferase, 2.3.1.61) and E3 (dihydrolipoyl dehydrogenase, EC 1.8.1.4) subunits, respectively [42,43]. Pretreatment with lipoamide protected cells against the detrimental effects of Zn^2+^ [40]. 

In the aging brain, an increased susceptibility of PDHC, NADP-isocitrate dehydrogenase (IDH-NADP), aconitase and glyceraldehyde-3-phosphate dehydrogenase to Zn and oxidative stress is linked with the presence of numerous cysteine–SH groups in their active centres, which require cytoprotective concentrations of glutathione, lipoamide and other free radical scavengers [14,106,107]. The synthesis of these species in aged brain decreases, whereas production of free radicals increases. As a result, several key enzymes of acetyl-CoA and energy metabolism are oxidatively modified [107]. This may be one of the causes of the increased incidence of Alzheimer’s disease and other neurodegenerative conditions in aged populations [106,108]. It would justify supplementation of those patients with N-acetyl-cysteine, tocopherol, glutathione or lipoamide as an adjuvant therapy [106,108,109].

One may assume that, in healthy neurons, intracellular Zn binds mainly with proteins requiring this cation for their biological activity. On the contrary, excessive amounts of accumulated Zn targeted other proteins, for which binding with Zn may be harmful (Figure 2). Half-maximal inhibition concentrations ([IC50]) of extracellular Zn against the PDHC and KDHC, aconitase and IDH-NADP inside undisrupted differentiated SN56 cells were equal to about 0.045, 0.033, 0.138 and 0.015 mmol/L, respectively [40,105,110]. In NC and DC SN56 cell lysates, Zn caused direct, similarly potent inhibition of PDHC, aconitase and IDH-NADP activities with [IC50] equal to about 0.08, 0.008 and 0.048 mmol/L, respectively [105]. On the other hand, Zn [IC50] values for DC-KDHC were found to be eight times lower than in NC-SN56 cells [110]. These [IC50] values lie within the low range of extracellular Zn concentrations in activated glutaminergic synapse, indicating that such a mechanism is pathophysiologically relevant [43,84]. Stronger inhibition of KDHC in DC-SN56 by Zn may contribute in part to their greater susceptibility to this cation due to a deeper suppression of metabolic flow through downstream steps of the TCA cycle [110] (Table 1). However, the mechanism of differential inhibition of KDHC by Zn in NC and DC SN56 remains unexplained. 

On the other hand, irreversible inhibition of aconitase by Zn resulted from competitive removal of iron from non-heme Fe-S domains in its active centre. In turn, IDH-NADP inhibition was evoked by accumulating free radicals [40,45,105,110,111]. Hence, the Zn-evoked excitotoxic impairment of neuronal cells results not only from the suppression of acetyl-CoA synthesis, but also from the limitation of its flow through the first three steps of the TCA cycle and the uncoupling of oxidative phosphorylations, yielding decreases in ATP levels and cell viability [112]. These effects resulted from the reciprocal facilitation of Zn and Ca influx from extracellular space through L, N and P/Q voltage-gated calcium channels (VGCC) and NMDA receptors [105,110,113]. These alterations were alleviated by L-type VGCC antagonist nifedipine, which decreased both Ca and Zn accumulation by 60–80%. N and P/Q type VGCC antagonists exerted a similar, but about two times weaker, protective effect due to a lesser inhibition of Zn and Ca accumulation in SN56 cells. As such, L-type VGCCs play a key role in Zn-evoked inhibition of pyruvate oxidation, followed by suppression of acetyl-CoA, ATP levels, ACh synthesis and loss of viability [105]. About 80% of Zn and Ca is accumulated in the cytoplasmic compartment, reflecting its larger volume in comparison to the mitochondria [110,112]. Nevertheless, several Zn transporters, including ZnT3, are involved in cation redistribution into the mitochondria and synaptic vesicles, causing Zn overload in these organelles in pathologic conditions [102]. Thus, Zn accumulated in cultured primary astrocytes aggravated lipopolysaccharide-induced NO synthesis, thereby exacerbating neuronal injury [14,114].

### 4.3. Thiamine Deficiency and Cholinergic Acetyl-CoA Metabolism

Thiamine takes part in vast number of energy-requiring functions in all tissues, including those in the peripheral and central nervous system [14,45,115]. Thiamine deficiency (TD) is a pathogenic condition that decreases acetyl-CoA synthesis and inhibits its metabolic flow through the TCA cycle in all tissues due to the inhibition of PDHC and KDHC E1 subunit activities [45,115,116]. For full activity, E1 subunits of these enzyme complexes require thiamine diphosphate (TDP), which is synthesised in the brain and other tissues by thiamine pyrophosphokinase (EC 2.7.6.2.) from ATP and thiamine supplied from diet. TDP deficiency is diagnosed in people who are chronically on thiamine-poor diets. Such conditions are identified—besides starving populations—most frequently in chronic alcoholics, in patients with AD, anorexia, malignancies, and in those on dialysis due to chronic renal failure or with various physical or mental disabilities [117]. TDP deficiency usually aggravates the course of basic disease. The brain is particularly susceptible to TDP deficiency due to high energy demands. TDP deficiency preferentially disturbs neurotransmission in the central and peripheral cholinergic neurons, yielding impairment of cognitive and motor functions, which are leading symptoms of this pathology [118]. 

In nerve terminals from the brains of rats with overt, pyrithiamine-induced TDP deficiency, both PDHC and KDHC activities in synaptosomal lysates and rates of in situ utilisation of pyruvate and ketoglutarate were significantly depressed due to a lack of proper saturation of E1 subunits with TDP. This indicates that TDP deficiency depressed enzyme expression and inhibited in situ activity. As a result, acetyl-CoA levels decreased in both mitochondrial and synaptoplasmic compartments of nerve terminals [119]. In turn, a decreased acetyl-CoA level in the synaptoplasm resulted from the inhibition of the ACLY pathway, yielding a 50% reduction in total ACh release and 90% depression of its quantal release. This indicates that ACLY-derived acetyl-CoA is almost exclusively responsible for the proper functioning of cholinergic transmission. In whole brain mitochondria originating predominantly from glial cells, TDP deficit inhibited metabolic fluxes through PDHC and KDHC steps in situ and enzyme activities in mitochondria, yielding a reduction in acetyl-CoA content [119,120]. Similar results were obtained with cholinergic SN56 neurons, in which amprolium, a competitive inhibitor of thiamine transport, caused TDP deficit with an inhibition of pyruvate oxidation resulting in the suppression of mitochondrial and cytoplasmic acetyl-CoA followed by significant loss of ACh synthesis, but no changes in ChAT (Figure 2). 

These data indicate that short-lasting TDP deficit exerts only functional inhibition of cholinergic neurons without affecting their structural integrity [121]. Hence, TDP deficit-evoked acetyl-CoA decrease in the mitochondrial compartments of neuronal and glial cells may be a primary signal impairing their functions, leading in a longer perspective to irreversible injury. They also may elucidate, in part, the mechanism of the beneficial effects of thiamine supplementation in traumatic brain injury in mice, which prevented inactivation of the KDHC and PDHC, thereby improving energy balance and reducing inflammatory reaction [122]. TDP deficiency is thought to be one of the conditions facilitating the onset and worsening the course of AD and other cholinergic encephalopathies due to its targeting of the same enzymes of energy metabolism as Aβ or Zn [48,116].

These conclusions were supported by studies of SN56 cholinergic NC and DC, in which 35% deficits of TDP were induced by amprolium. This inhibitor of thiamine uptake caused decreases in PDHC activity and the acetyl-CoA level in both cell phenotypes. However, these conditions caused no change in ChAT activities, but they evoked a marked decrease in ACh synthesis in DC but not in NC [45]. Such differential effects could result from the fact that the rate of ACh synthesis in DC was over three times higher than in NC and required a higher supply of acetyl-CoA. However, a marginal 20% TDP deficit usually remains silent and does not present overt clinical symptoms [48,116]. Moreover, DC SN56 cholinergic cells grown in thiamine-deficient medium presented no significant changes in their PDHC activity, acetyl-CoA, ACh synthesis or viability against cells grown in thiamine-supplemented medium. A marginal excess of Zn (free Zn^2+^ 0.01 mmol/L) caused no significant alterations in these parameters against cholinergic SN56 DC grown in thiamine-supplemented medium [39,40,45] (Table 1). On the other hand, thiamine-deficient SN56 cells appeared to be highly susceptible to low Zn^2+^ concentrations, which caused significant inhibition of PDHC, aconitase, NADP-ICDH and KDHC activities, suppression of acetyl-CoA, ACh and TDP levels, and loss of viability. This was due to the fact that TDP deficit and Zn excess simultaneously targeted the E1 and E2/E3 subunits of the PDHC and KDHC, respectively, yielding reciprocal aggravation of single inhibitory effects [39,45]. There was a significant, direct correlation between TDP content, Ach level and viability of SN56 cells. On the other hand, TDP-deficient astroglial C6 cells were several times more resistant to Zn excess than SN56 neuronal cells despite the accumulation of larger amounts of this cation. Thus, astroglial cells retained functional capacity in conditions already harmful for neuronal cells [39,45]. As such, SN56 neuronal cells subjected to TDP deficit may be rescued from excitotoxic Zn insult by co-cultured astroglial cells that take up Zn excess from their vicinity. In effect, the accumulation of Zn in SN56 decreased, yielding an increase in TDP content and the restoration of all parameters of acetyl-CoA metabolism back to sub-control values [39] (Table 1). Such pro-cholinergic and neuroprotective properties of astroglia remain in accord with their physiological role as a scavenger of Zn excess and other metals from extracellular compartments of the brain [104].

### 4.4. Nerve Growth Factor (NGF) and Acetyl-CoA in Cholinergic Cells 

Cholinergic neurons of the basal forebrain are responsible for several cognitive functions of the brain; they include learning, memory updating, extinction and renewal, behaviours, attention, addiction and fear–reward-related processes. NGF is the key cholinotrophic growth factor that regulates the maturation, development and maintenance of cholinergic basal forebrain neurons (cBF) during embryonal and postnatal life. It stimulates gradient-guided axonal growth of the cholinergic basal forebrain neurons (cBF) in the embryonal and early postnatal periods through unspecific p75NTR receptors activating NFκB signalling. On the other hand, activation of the p75NTR-JNK signal transduction pathway promoted apoptotic neuronal death [123]. Therefore, the balance between these two p75NTR-linked intracellular signalling pathways is important for the proper developmental formation of targeting cholinergic pathways. The expression of high-affinity NGF-specific TrkA receptors in the mouse brain appeared on the 18th embryonal day and increased during postnatal maturation, stimulating increases in ChAT, VAChT, M2 receptors and ACh synthesis in cholinergic neurons of basal nuclei and their innervated cortical regions [79,122,124]. Distribution of NGF and TrkA receptors correlated well with the density of cholinergic ChAT and VAChT immunopositive neurons, indicating its tight functional link with cholinergic neurons [125]. Hence, in the embryonal period, with minimal expression of TrkA receptors and low cholinergic phenotype, p75NTR exerted positive effects on the development of cBF neurons. On the other hand, with a high expression of cholinergic phenotype in mature brain, increased levels of p75NTRs started mediating programmed cell death and the downregulation of cholinergic phenotype expression [23,123,126]. One of the mechanisms of increased vulnerability of NGF-stimulated cholinergic neurons to common neurotoxins could be the increased competition for acetyl-CoA between the pathway of ACh synthesis and the TCA cycle [14]. A high level of p75NTR in differentiated cells could also promote the formation of p75NTR homodimers that, after binding with NGF, conveyed ceramide-evoked programmed cell death [127].

This hypothesis is supported by studies of NGF effects on acetyl-CoA metabolism in SN56 cholinergic neurons, with differential expressions of p75 NTR, TrkA receptors and cholinergic phenotype [99,101] (Table 1). As such, native NC SN56 cholinergic cells in culture displayed only p75NTR and did not respond to NGF treatment, which is compatible with the absence of TrkA receptors in embryonal cBF [121]. Conversely, treatment with cAMP + retinoic acid of both native TrkA(-) p75NTR(+) and transgenic TrkA(+)p75NGF(+) cholinergic neuronal cells caused their similar morphologic maturation, several-fold increases in cholinergic phenotypes and density of p75NTR, along with a shift of acetyl-CoA from the mitochondrial to cytoplasmic compartments, respectively [98,126]. The latter shift was compatible with an increased demand of differentiated cells for cytoplasmic acetyl-CoA supply for activated ACh synthesis and release [98,126] (Table 1). However, the addition of NGF to cAMP/RA-differentiated SN56 caused the suppression of their cholinergic parameters and acetyl-CoA levels and aggravated the cholinotoxic effects of Aβ and NO excess. These negative effects were alleviated by the simultaneous addition of anti-p75NTR antibody [98,126] (Table 1). This indicates that both the negative and positive effects of NGF interactions with p75NTR could be mediated through respective shifts and alterations in mitochondrial/cytoplasmic concentrations of acetyl-CoA in septal cholinergic neurons [14].

Such findings are compatible with the results of recent studies on whole mice [23], which revealed that MCAO followed by 45 min whole-body hypoxia caused a loss of neurons in the ipsilateral cortex and thalamus. Two weeks after insult, suppression of pyruvate metabolism, decreases in acetyl-CoA, acetylcholine and serotonin and reduced glutathione levels, but no alterations in dopamine and glutamate levels, were observed [23] (Table 1). Post-surgery, animals treated with oral administration of a p75NTR modulator, LM1 1A-31 ([2-amino-3-methyl-pentanoic acid (2-morpholin-4-yl-ethyl)-amide]), showed improved recovery of motor and cognitive abilities, restored acetylcholine and other neurotransmitter levels, as well as reduced brain atrophy, neurodegeneration, tau pathology and microglial activation [23]. Moreover, in vivo knockdown of p75NTR with antisense oligonucleotides brought about increased ChAT activity and ACh release in the hippocampus of mature rats [128]. These data indicate that suppression of the p75NTR signalling pathway may exert a beneficial effect on the outcome of several neurodegenerative diseases through the alleviation of disturbances in energy and acetyl-Co-A metabolism [14,23,129]. 

### 4.5. Citrate and ACLY Key Players in Cholinergic Acetyl-CoA Metabolism

Over 90% of intramitochondrial acetyl-CoA derived from pyruvate (glucose) is converted to citrate via citrate synthase, which is utilised mainly in the TCA cycle. Some amounts of citrate are transported to the neuronal cytoplasm through the SLC25A1 citrate/malate antiporter located in the inner mitochondrial membrane [130]. In the cytoplasm, in the presence of ATP and CoA-SH, citrate is converted back to acetyl-CoA by ACLY to be utilised in the acetylation of a vast number of cytoplasmic and nuclear proteins, peptides and lipids [16,131]. A fractional contribution of citrate to the cytoplasmic pool of acetyl-CoA depends on cell type and the brain region. Studies with a strong and specific inhibitor of ACLY, (-) hydroxycitrate (HC, Ki 3.8 µM), demonstrated that, in brains of suckling animals, 1 mmol/L of the inhibitor brought about 60% suppression of fatty acid synthesis in glial cells [130]. Such strong inhibition of fatty acid synthesis by HC in the myelinating brain may result from a prevalence of ACLY over the NAA pathway in providing acetyl groups for myelin lipid synthesis in maturing oligodendroglia [132,133,134]. It should be emphasised that, at such concentration, HC did not inhibit aconitase, CS, IDH-NADP, ChAT or other enzymes involved in acetyl-CoA and energy metabolism [134,135].

On the other hand, in the adult rat brain, 2.5 mM HC resulted in 17, 30 and 55% inhibition of ACh synthesis in the hippocampus, caudate nucleus and septum, respectively [136]. These differences may result from the existence of subpopulations of cholinergic neurons with variable fractional contributions of citrate-derived acetyl-CoA to ACh synthesis in a particular brain region. The provision of acetyl-CoA for ACh synthesis through ACLY is facilitated by preferential localisation of this enzyme in cholinergic neurons. There are highly significant, positive correlations between ChAT and ACL activities in nerve terminal fractions isolated from regions of variable density of cholinergic innervation. Activities of ACLY in cholinergic terminals calculated from a ChAT vs. ACLY correlation plot were estimated to be as high as 40 nmols/min/mg protein, whereas in non-cholinergic ones, they were equal to 2–4 nmols/min/mg protein [137]. These findings were confirmed by in vivo studies showing significant, over 30% decreases in ACL activity accompanying 70–80% losses of ChAT in synaptosomes from the hippocampus and cortex after electrolytic lesions of septum or 192 IgG-saporin immune lesions of cholinergic neurons, respectively [138]. The tight link between ACLY and the cholinergic system was confirmed by the existence of a BNIP-H-induced formation of an ACLY-ChAT complex, which stimulates cholinergic neuritogenesis and neurotransmission [139] (Figure 2).

ACLY is activated by phosphorylation of its serine455 residue [140]. In SN56 cholinergic neurons, dephosphorylation of ACLY by overexpression of protein histidine phosphatase (PHP) inhibited enzyme activity, resulting in a 30% reduction in ACh content and 10% loss of cell viability. Suppression of PHP by specific siRNA restored ACh content and cell viability [141,142]. These data indicate that the fractional contribution of the ACLY pathway to the cytoplasmic acetyl-CoA pool for ACh synthesis in cholinergic neurons is lower than that for fatty acid synthesis in oligodendrocytes [133,136,143]. It is possible that the majority of acetyl-CoA in depolarised cholinergic terminals is supplied to the cytoplasmic compartment directly by Ca-sensitive permeability transition pores (PTPs) in the mitochondrial inner membranes [19,79]. Streptozotocin-induced diabetes increased ACLY activity in mouse hippocampus and brain cortex but stimulated metabolic flux of oxaloacetate to lactate only in the hippocampus, indicating the differential adaptation of brain regions to hyperglycaemia [140]. Accordingly, in streptozotocin-diabetic rats, increases in pyruvate oxidation, acetyl-CoA level and ACh synthesis were observed in the brain nerve terminal fraction [143]. These data indicate that ACLY plays a key role in feeding both mitochondrial oxidation and cytoplasmic ACh synthesis in diabetic cholinergic nerve terminals. 

Moreover, extracellular citrate may serve as a source of acetyl-CoA in neuronal cells. The concentration of citrate in blood is about 0.16 mmol/L [144], whereas the Km value for the citrate transporter on the BBB is around 0.6 mmol/L; this precludes the abundant transport of citrate from circulation into the brain under normal conditions. This indicates that the majority of citrate in brain extracellular space originates from its own internal metabolism. The citrate level in this compartment is in the range of 0.25–0.35 mmol/L [145,146]. Such concentration of citrate is sufficient for its uptake by neuronal plasma membranes using the sodium-dependent citrate transporter encoded by the SLC13A5 gene [147]. The Km for citrate in the SLC13A5 transporter located on neuronal plasma membranes was estimated to be in the range of 0.3 mmol/L. This indicates that rates of citrate uptake by neurons may depend on alterations in its concentration in the brain interstitial compartment [148]. The main net producers of extracellular lactate and citrate in the brain are astrocytes [4]. They display relatively low rates of citrate utilisation in oxidative metabolism in mitochondria and its cleavage by ACLY in the cytoplasmic compartment [149]. In effect, the rate of citrate release from astrocytes was reported to be 12 times higher than from neurons [150]. Therefore, after lactate, the citrate may constitute a secondary, although important, line of metabolic flow from astrocytes to the neurons [5]. Several studies have recently focused on ACLY as a central metabolic enzyme in cancer [151]. A number of inhibitors targeting SCL13A5 and ACLY have been developed as potential therapeutic agents against different malignant tumours of the brain and peripheral tissues, limiting cytoplasmic acetyl-CoA synthesis [148,152] (Table 1).

Nevertheless, mutations in the SLC13A5 gene cause impairment of brain development in humans and experimental animals, pointing to the significance of exogenous citrate for acetyl-CoA metabolism [153]. The exogenous citrate is predominately metabolised in the cytoplasm by ACLY, yielding acetyl-CoA mainly used for fatty acid and cholesterol synthesis in the cytoplasm of neuronal and oligodendroglial glial cells. This pathway is activated under nutrient–limiting conditions, such as hypoxia or hypoglycaemia [148]. However, endogenous citrate produced by citrate synthase reaction in the mitochondria still remains a main fraction of acetyl-CoA utilised in the TCA cycle and feeding the cytoplasmic pool of this metabolite. This is evidenced by a relatively small (10%) dilution of labelled [13C]acetyl-CoA derived from [U-13C]-glucose by unlabelled extracellular citrate [150]. Nevertheless, there are no data indicating the existence of functionally separate endogenous and exogenous pools of this metabolite in the cytoplasm. On the other hand, rare, negative mutations of the SLC13A5 gene cause autosomal recessive epilepsy, pointing to the key significance of this substrate in the presumable deficit of acetyl-CoA for ACh synthesis and other neurotransmitter functions in the brain. Accordingly, elevated citrate levels in CSF were reported in those patients and in mice with the SLC13A5 gene deletion, in which low citrate levels were also recorded in the para-hippocampal cortex [154,155,156].

## 5. Compartmentation of Brain N-Acetylaspartate and Acetyl-CoA Metabolism

It seems that, irrespective of the precursor, neuronal intramitochondrial acetyl-CoA constitutes a uniform metabolic pool without discrete subdomains, available either for catabolic or anabolic pathways. Over 90% of acetyl-CoA is utilised for energy production in the TCA cycle linked with the respiratory chain. Another 1–3% of acetyl-CoA is utilised by aspartate N-acetyltransferase (NAT, EC 2.3.1.17) located exclusively in neuronal mitochondria for N-acetylaspartate (NAA) synthesis [11,83,157]. Therefore, over 95% of whole brain NAA is concentrated in neurons as the site of its synthesis. One could estimate that, at a 10 mmol/L level of NAA in the whole brain, its real concentration in the neuronal compartment may be several times higher than in the neuroglia [156]. The rate of NAT reaction strongly depends on acetyl-CoA concentration in neuronal mitochondria. The acetyl-CoA concentration in this compartment is about 12 µmol/L, whereas its Km is equal to 58 µmol/L [14]. Therefore, inhibition of neuronal PDHC by Zn or other inhibitors reduced the rate of acetyl-CoA synthesis, causing secondary decreases in NAT in situ activity and NAA level, respectively [11]. Differentiation caused a 30% increase in AAT level but resulted in a 25% decrease in NAA content. This apparent divergence may be explained by a differentiation-evoked 60% decrease in mitochondrial acetyl-CoA [11,97]. On the other hand, aspartate concentration in the brain is equal to 3500 µmol/L saturated NAT, with Km for this substrate being equal to 580 µmol/L [158,159]. Therefore, in vivo, at saturating levels of L-aspartate, the yield of an NAT reaction may depend almost exclusively on acetyl-CoA concentration in the neuronal mitochondrial compartment. For instance, Zn in a low-toxicity concentration caused similar 50% decreases of pyruvate oxidation as well as acetyl-CoA and NAA levels in SN56 neuronal cells [11]. This resulted in Zn concentration-dependent decreases in NAA levels strongly correlating with reductions in mitochondrial acetyl-CoA [11]. 

This confirms, for the first time, the hypothesis that changes in NAA level seen in MRI imaging of a neurodegenerating brain may correspond with alterations of acetyl-CoA in the neuronal mitochondrial compartment [11,15,83,95,157]; in fact, 85% of SN56 neuronal NAA was located in their mitochondria [160]. After release from neurons, NAA is transported through myelin gaps to oligodendrocytes by Na^+^-dependent high-affinity dicarboxylate transporter, whereby aspartoacetylase (N-aspartyl aminoacylase EC3.5.1.15, ASPA) is cleaved to acetate, which is converted back by cytoplasmic ACSS1 to acetyl-CoA and utilised for fatty acid/cholesterol synthesis necessary for myelin formation [132]. Aspartate is used for energy production via malate–aspartate shuttle, which supports its own oligodendroglial energy metabolism [133,161]. Post-mortem studies of MS brains revealed low levels of NAA and acetate in the cortex and white matter, respectively. It reflected low NAA levels in insulted neurons, which slowed down the transfer of this metabolite to oligodendroglia, yielding deficits of acetate and acetyl-CoA for lipid synthesis [162] (Figure 2).

In addition, no NAA or aspartate recycling from oligodendroglia to neurons was reported. This would be compatible with some NMR studies that suggest oligodendroglia as an NAA-enriched compartment. This finding infers that a proper NAA signal in NMR may reflect the competence of both neuronal and oligodendroglial cells [163,164]. On the other hand, a marked increase in NAA may be harmful for oligodendrocytes. An inherited defect of acetylaspartate deacylase, known as Canavan disease, demonstrates a phenotype of congenital juvenile leukodystrophy. Demyelination results from deficits of acetate to be converted to acetyl-CoA and used for myelin lipid synthesis. However, genetic suppression or chemical inhibition of NAT activity by specific inhibitors caused a decrease in NAA concentration, prevented the loss of neurons and alleviated some clinical symptoms of the disease, apparently without improving acetyl-CoA synthesis from acetate [165,166,167]. This indicates that NAA excess rather than acetyl-CoA deficit may be a more important pathogenic factor in this disease. Such a claim was also supported by the finding that acetyl-CoA synthesis from pyruvate is unnecessary for myelin preservation [57]. However, overexpression of NAT (Natl8) in mice brain neurons increased NAA levels but caused no neurological impairments, indicating that increased levels of NAA in the brain are not cytotoxic per se [167]. Conversely, depletion of the ASPA gene in oligodendrocytes brought about Canavan-like neurological disturbances. In such conditions, intracranial ASPA gene replacement therapy targeting oligodendrocytes in ASPA-deficient mice resulted in regression of this pathology. These findings may pave the path to gene therapies for this and other leukodystrophies [168]. There are also close interactions between the loss of mitochondrial metabolism and cholinergic neuron degeneration in APP/PS1 AD mice, which, at the age of 6–8 months, displayed Aβ deposits and cognitive deficits accompanied by losses in cholinergic neurons, ChAT, VAChT, choline and NAT levels in the hippocampus and medial septum. Activation of cholinergic circuits in 4-month-old APP/PS1 AD mice with injections of M3dq virus followed by clozapine-N-oxide treatment prevented losses in cholinergic neurons and NAA levels [169]. On the other hand, activities of ChAT and levels of NAA displayed an inverse correlation in the septal and cerebellar regions of the rat brain, containing high and low densities of cholinergic neurons, respectively [136,159]. This indicates that, in cholinergic neurons, NAA levels may be lower than in the noncholinergic ones due to competition for acetyl-CoA, which is preferentially released to the cytoplasm in the former [98]. It also remains in accord with data demonstrating a decrease in NAA level in DC against NC SN56 cholinergic neuronal cells [11] (Figure 2).

Alterations in acetyl-CoA availability in neuronal mitochondria could also exert a proportional transcellular effect on its metabolism in oligodendroglial cells. Namely, the limitation of acetyl-CoA synthesis in the neuronal compartment of an aging brain due to inhibition of the PDHC from age activated PDK3 and impaired supportive interactions with other brain cells, including oligodendrocytes, making them more vulnerable to several harmful signals [80,170]. In response, oligodendroglia may also provide fewer precursor substrates of acetyl-CoA to neuronal axons, reciprocally affecting their integrity [72,73]. These data are in accord with our findings on the direct correlations of NAA and ACh levels against PDHC activities and acetyl-CoA content in SN56 cholinergic cells subjected to Zn excess or TPP deficits [11,45]. 

## 6. The Contribution of Acetate to Brain Acetyl-CoA Metabolism

Endogenous acetate is produced by the deacetylation of several acetylated proteins, histones and low-molecular-weight compounds by multiple deacetylases or from acetyl-CoA by acetyl-CoA hydrolase (EC3.1.2.1). In order to enter any metabolic pathway, acetate has to be reactivated by acetyl-CoA synthases (ACSS) 1 or 2 (EC 6.2.1.1.) located in cytoplasmic or mitochondrial and nuclear compartments, respectively [171]. ACSS activity in neurons is about two times higher than in astrocytes; however, astrocytes take up and metabolise extracellular acetate to CO_2_ 18 times faster than nerve terminals, due to a higher expression of monocarboxylate transporters on their plasma membranes [172,173]. On the other hand, oligodendrocytes take up NAA released by neurons, which is then hydrolysed by cytoplasmic ASPA, yielding endogenous acetate that is reconverted to acetyl-CoA by ACSS1. Thereby, these two classes of neuroglia constitute the main acetate-utilising compartments in the brain [173]. Oligodendrocytes utilise acetate converted to acetyl-CoA mainly for fatty acid and cholesterol synthesis, which is necessary for the formation of myelin sheets, thereby supporting myelinisation and growth of axons [174,175]. 

Adequate provision of acetyl-CoA for fatty acid and cholesterol synthesis is also indispensable for the maintenance of structural and functional integrity of plasma membranes and intracellular structures in all brain cells. Recent data demonstrate that cytoplasmic fatty acid synthesis in SAMP8 rapidly aging mice is affected by a low acetyl-CoA level in neuronal cells [174,175]. Carboxylation of acetyl-CoA to malonyl-CoA by cytoplasmic acetyl-CoA carboxylase (ACC-1, EC 6.4.1.2.) constitutes the first, rate-limiting step in the synthesis of structural fatty acid and lipids in neurons and glial cells. This reaction was found to be an important factor decreasing the level of acetyl-CoA in cultured primary neurons, clonal HT-22 hippocampal neuronal cells, as well as in hippocampal and cortex slices from rapidly aging SAMP8 mice [10,175]. Derivatives of the flavonoids curcumin and fisetin (J147, CMS121, CAD031) were activated by phosphorylation of 5′AMP-activated protein kinase (AMPK), which in turn phosphorylated ACC-1 at serine 79. This caused the inhibition of ACC-1 activity and diminished the utilisation of cytoplasmic acetyl-CoA for fatty acid synthesis. This resulted in the elevation of acetyl-CoA content, which brought about longer survival of cultured neuronal cells (Table 1). Several months feeding of rapidly aging SAMP8 or APPswe/PS1dE9 double-transgenic AD mice with fisetin prevented development of age-evoked cognitive and locomotor deficits at the 13th month of life [174]. These anti-senescent properties of fisetin may also result from its ability to alleviate decreases in structural, synapse-associated proteins such as Homer, Arc, pro-stress SAP102, HSP40 or HSP90, as well as suppression of age-elevated pro-stress HSP60. It also reduced astrogliosis and levels of inflammation-associated p25/35, P-JNK/JNK proteins, as well as suppressed synthesis of several prooxidant derivatives of arachidonic acid, docosahexaenoic acid and linoleic acid. These properties of fisetin prevented age-disrupted neuronal homeostasis in the hippocampus of 13-month-old SAMP8 mice [174]. Similar neuroprotective effects were observed in cell culture by direct specific, competitive inhibition of ACC-1 with 5-(tetradecyloxy)-fluoric acid (TOFA) or by suppression of the enzyme expression by CTsiRNA [175]. An increased level of cytoplasmic acetyl-CoA facilitated its transport to the nucleus and stimulated histone acetyltransferase, which enhanced acetylation of histone H3K9, resulting in the improvement of cognitive status of SAMP8 mice [174,175] (Table 1).

Neurons express cytoplasmic acyl-CoA synthase short-chain family member (ACSS2), which synthesises acetyl-CoA from acetate to be used for lipid synthesis. Experimental subarachnoid haemorrhage in mice caused a marked decrease in ACSS2 levels and negative outcome caused by autophagy and brain oedema [176]. Transfection with ACSS2-lentivirus vector increased the ACSS2 level along with the downregulation of pro-apoptotic and upregulation of anti-apoptotic proteins both within in vivo and primary neuronal culture [176]. It might be hypothesised that increases in cytoplasmic and nuclear acetyl-CoA levels could be involved in this neuroprotective mechanism, mediated by alterations on histone acetylations [14,15,177,178] (Table 1).

Exogenous acetate, through ACSS2, may also provide acetyl-CoA, which corrected deficits of this metabolite in neurons from the brains of heterozygotic mice with partial PDHC deficiency. Brain slices expressing low levels of PDHC contained lower glutamate and acetyl-CoA concentrations than the WT ones. However, they also displayed an increase in glutamatergic excitatory postsynaptic currents (sEPSC) [54]. This could result from excessive release of glutamate from energy-deficient over-depolarised glutamatergic terminals and impairment of its clearance from the synaptic cleft by adjacent depolarised astrocytes and neurons. The latter also displayed acetyl-CoA shortages, yielding TCA cycle inhibition and ATP deficits. The addition of 5 mmol/L acetate normalised the acetyl-CoA level in brain slices from PDHC-deficient mice, yielding an improvement of their glutamatergic metabolism and neurotransmission. Systemic application of [1,6^13^C]glucose and [1,2^13^C]acetate revealed that the latter is the preferred precursor of acetyl-CoA in PDHC-deficient mice brain cortex slices [54]. Moreover, acetate used in pathophysiologically irrelevant 100 mmol/L concentrations increased HT-22 cell viability, apparently due to the elevation in their acetyl-CoA level [175]. On the other hand, in the control animals, glucose was the preferred precursor of acetyl-CoA. These data indicate that acetate may become a significant alternative source of acetyl-CoA in brains with inherited or acquired PDHC deficits. As such, acetate may present therapeutic potential for clinical interventions in various neuropathological conditions evoked by lowered PDHC activities.

Other studies, however, revealed that inhibition of ACC-1 and fatty acid synthesis by cerulenin or TOFA increased the death rate under glucose/oxygen deprivation conditions in cultured SH-SY5Y neuroblastoma and primary brain cortex cells. In addition, in the in vivo middle carotid artery occlusion stroke model, cerulenin increased infarct size, inflammatory reaction of microglia and memory deficits [12]. Such a finding would be compatible with the fact that fatty acids and cholesterol are indispensable for the maintenance of structural integrity of all brain cell types, including neurons. The source of these discrepancies remains unclear, as the acetyl-CoA level was not assessed in this study. Nevertheless, both reports reveal the existence of a reciprocal interdependence between the utilisation of acetyl-CoA for mitochondrial energy production and its utilisation for cytoplasmic fatty acid synthesis. They indicate that maintenance of the balance between mitochondrial and cytoplasmic pools of acetyl-CoA is a significant factor for neuronal functions and viability [14].

Conversely, Dong and Brewer [178] presented the opposite results concerning mechanisms linked with acetate-acetyl-CoA-dependent neuroprotection, demonstrating that aging results in a marked 2–2.5-fold increase in whole brain acetyl-CoA levels in both 3x TgAD and WT mice, irrespective of gender. Moreover, acetyl-CoA levels in male 3xTgAD appeared to be over two times higher than in age-matched WTs (Table 1). Aging brains displayed depression in citrate and aconitate levels, suggesting inhibition of acetyl-CoA utilisation in the TCA cycle, likely at the citrate synthase step. However, the mechanism of this apparent inhibition remains obscure. In fact, the activity of CS in the hippocampus and septum of 24-month-old rats was decreased by 45 and 70% compared to the young controls, respectively [179] (Table 1). The activity of CS in the brain is one to two orders of magnitude higher than the activities of PDHC and other enzymes of the TCA cycle [180]. In addition, affinities of brain CS to acetyl-CoA and oxaloacetate are high, being equal to 4.5 and 5.0 µM, respectively [181]. This minimalizes the possibility of limitation of acetyl-CoA utilisation by age-related decreases in CS levels. Causes of this inconsistency remain unexplained. Nevertheless, the age-evoked increase in fatty acid β-oxidation could compensate energy deficits evoked by PDHC suppression and contribute to increased acetyl-CoA levels in brains of both WT and Tg mice [179]. On the other hand, these data did not indicate in which cellular compartment(s) these alterations took place. 

Aging and Aβ overload evoke several metabolic disturbances in the brain, including decreases in PDHC activity and acetyl-CoA level [60,62]. Sphingosine kinase 1 (SphK1) is an acetyl-CoA-activated enzyme that synthesises N-acetyl-sphingosine (NAS) from acetyl-CoA and sphingosine. In turn, NAS acetylated SN65 and activated COX2, which stimulated synthesis of several Specialised Pro-resolving Mediators (SPMs) from arachidonic and docosahexaenoic acids. Compounds such as Lx4A, eicosapentaenoic acid and RvD1 resolved neuroinflammation caused by Aβ and diverse neurotoxic insults, thereby improving memory in aged rats [182]. On the other hand, decreased levels of mRNA and activity of SphK1 were found in APP/PS1 AD mice neurons, but not in astroglial and microglial cells. This was accompanied by significant increases in proinflammatory and decreases in anti-inflammatory cytokines and the appearance of memory deficits in 12-month-old animals. These negative alterations were alleviated in APP/PS1-SphK1Tg mice. A study of isolated SphK1 revealed that its acetylation by exogenous acetyl-CoA has a Km equal to 58 µmol/L [182]. Additionally, purified COX2 was acetylated only in the presence of both SphK1 and acetyl-CoA. These findings remain in accord with the decreased level of SphK1 in brain neurons of AD patients [182]. Neuronal acetyl-CoA level is several times lower than its Km against SphK1 [14]. Therefore, the yield of this pathway strongly depends on acetyl-CoA availability, which is reduced in AD brains. These data indicate that SphK1 could be a potential target for therapeutic approaches to AD and other neuroinflammatory conditions, which should consider improvement of acetyl-CoA metabolism [183]. Namely, Aβ1-42 in a high 10 µmol/L concentration caused a decrease in the acetyl-CoA level in APP/PS1 mice microglia but not in neurons (Table 1). On the contrary, it suppressed SphK1 activity in neurons but not in microglia [183], despite the fact that NAS and acetylated COX2 were decreased by Aβ both in neurons and microglia. However, for normalisation of the NAS level, neurons required SphK1, whereas in microglia, and addition of acetyl-CoA was required. This indicates that different mechanisms contributed to SPM-deficits-evoked degeneration of neuronal and microglial cells. 

## 7. Fatty Acids and Ketone Bodies in Brain Acetyl-CoA Metabolism

Beta-hydroxybutyrate and acetoacetate (BHB) are the products of fatty acid oxidation in the liver. Under normal conditions, BHB concentration in the blood is in the range of 0.05 mmol/L, which excludes its effective transport into the brain. Pyruvate at similar concentrations is transported five times faster due to ten times higher affinity to MCT2 [25]. However, during starvation, stressful conditions, high-fat diet, diabetic ketoacidosis or hypoxia, BHB is synthesised and released from the liver in marked amounts that may reach blood concentrations as high as 5–20 mmol/L. Under such conditions, BHB is transported through the blood–brain barrier and enters both neuronal and oligodendroglial cells through MCT2 at Km 1.0 mmol/L, and is then transported to the mitochondria [24,184]. BHB is metabolised to acetyl-CoA by the β-hydroxybutyrate dehydrogenase (HBDH, EC 1.1.1.30), 3-oxoacid CoA transferase (EC 2.8.3.5.) and acetoacetyl-CoA thiolase (EC 2.3.1.9) pathways. In ketonemic conditions, BHB may supply up to 30% of neuronal acetyl-CoA complementary to the pool derived from glucose [14,21]. In the brain, only astrocytes express a high level of carnitine palmitoyltransferase and oxidise fatty acids via β-oxidation generating endogenous ketones, but they are not capable of metabolising them [185]. For this reason, ketones released from the astroglia are utilised by neighbouring neurons, yielding acetyl-CoA supporting energy metabolism and ACh synthesis [186,187]. Intermittent metabolic switching of animals from fasting–exercises-induced ketonemia to resting and feeding non-ketonemic conditions increased neuronal plasticity and brain resilience to injury and stressful conditions and improved cognitive functions [184] (Figure 1 and Figure 2). This may be explained by the simultaneous upregulation of alternative pathways for acetyl-CoA synthesis. Long-term hyperglycaemia-ketonemia in streptozotocin-diabetic rats increased BHB utilisation and acetyl-CoA synthesis in the whole brain mitochondria, as well as acetyl-CoA synthesis in the synaptosomal mitochondria and ACh synthesis in the synaptosomal cytoplasm [143] (Table 1).

Fasting ketonemia may also exert protective effects on brain cells in different pathologic conditions. Susceptibility of rat brain neurons to anoxia, induced by cardiac arrest lasting 8 min, was alleviated by long- or short-term dietary restriction-evoked ketonemia preceding this insult. The calorie-restricted control group displayed higher levels of BHB in the brain but no changes in acetyl-CoA and ATP concentrations against controls fed ad libitum [188]. However, the fasted group displayed higher levels of ATP and acetyl-CoA in their brains after cardiac arrest, which improved survival and neurological outcomes when compared to the non-starved controls [188]. This finding demonstrates that ketone bodies may serve as an alternative source of acetyl-CoA to support neuronal integrity under conditions limiting its provision from glucose-derived pyruvate.

Moreover, clinical studies of elderly patients with mild cognitive impairment revealed that ketonemia induced by caloric restrictions caused some improvements in their cognitive functions [189,190]. Accordingly, PET/MRI studies of AD patients revealed that AcAc/BHB metabolism in their brains did not change with the coexisting suppression of glucose uptake [191]. Meta-analysis of eight clinical protocols with AD patients confirmed the efficacy of therapeutic dietary ketosis for improvement in their cognitive status [190].

Acute stress induced by immobilisation followed by bright light exposure caused an increase in BHB levels in the blood and acute increase in HBDH expression, resulting in 100% elevation of the acetyl-CoA level in the prefrontal cortex of mice brains [192] (Table 1). No such alterations took place in the hippocampus. This indicates that the synthesis of acetyl-CoA may be quickly increased in the prefrontal cortex as a protective response to stressful conditions [192]. In this manner, acute systemic ketonemia would improve neuronal performance in relation to survival in threatening situations due to the increased synthesis of acetyl-CoA fraction independent of the PDHC. Moreover, in the brain synaptosomes from streptozotocin-diabetic rats, a moderate 2.5 mmol/L BHB concentration suppressed oxidation of 2.5 mmol/L pyruvate but did not alter acetyl-CoA levels and ACh synthesis [143] (Table 1). BHB alone was able to maintain acetyl-CoA and ACh levels equal to 25% of that in the presence of pyruvate alone. These data indicate that ketone bodies may supplement pyruvate in the provision of acetyl-CoA for ACh synthesis. 

Medium-chain C8-C10 fatty acids (MCFAs) are effectively taken up from circulation by the brain and provide acetyl-CoA through the sequence of reactions started by mitochondrial medium-chain acyl-CoA dehydrogenase. However, neither neurons nor oligodendroglia effectively utilise these energy substrates. Astroglia form the main cellular compartment utilising both MCFAs and long-chain fatty acids (LCFAs) in the brain [193]. Astrocytes efficiently take up MCFAs and LCFAs from extracellular space and transport them directly to mitochondria without the participation of carnitine acyltransferases [192]. Experiments with [U^13^C]C8-10 revealed that [^13^C]acetyl-CoA, derived from their beta-oxidation is preferentially incorporated into citrate, which may increase the rate of the TCA cycle. This increases α-ketoglutarate synthesis in astrocytes and its conversion to glutamate and glutamine by glutamate dehydrogenase/AST and glutamine synthetase reactions, respectively [194,195]. Increased amounts of glutamine are then exported from astrocytes to glutamatergic and GABA-ergic neurons to support their energy and neurotransmitter synthesis. These data are in accord with the thesis that acetyl-CoA derived from MCFA oxidation in astroglia is a primary signal stimulating the cooperative transfer of glutamine from these cells to neurons. They are compatible with observations that C8-C10 supplementation improved cognitive functions in patients with Alzheimer’s and Huntington’s disease and in AD/HD models of transgenic mice [194]. However, there are no direct data that would evidence the medium-chain fatty acid oxidation increasing the acetyl-CoA level in astroglia and particularly in their mitochondria [194]. Other experiments carried out on the hippocampal immortalised HT22 line demonstrated that both BHB and decanoic acid may increase the gene expression, amount of protein and the activities of Sirtuins that, by deacetylation, activated long-chain acyl-CoA- dehydrogenase and several mitochondrial proteins, including respiratory chain complexes [196]. They apparently increased acetyl-CoA production and utilisation in energy-producing pathways, although no direct assessment of this metabolite has been performed. Our studies on cultured SN56 neurons are compatible with these findings, revealing that a high level of acetyl-CoA in the mitochondrial compartment warrants proper rates of ATP synthesis and maintenance of cell viability [14,105]. Conversely, diverse PDHC inhibitors, such as Zn, Aβ, nitrosyl radicals and TDPD, decreased acetyl-CoA content in clonal SN56 cholinergic neuronal cells, nerve terminals and whole brain mitochondria, affecting their viability [14,105,119]. 

## 8. Endoplasmic Reticulum and Nuclear Pools of Acetyl-CoA 

The nucleus and endoplasmic reticulum are involved in gene expression, protein processing and their post-translational modifications, including secretory and autolytic pathways, respectively. In both compartments, multiple acetylation and deacetylation reactions, along with several other post-translational modifications of proteins, adjust their properties to cellular requirements.

### 8.1. Endoplasmic Reticulum Acetyl-CoA

Except for the large cytoplasmic pool of acetyl-CoA, there are smaller, yet functionally important pools of extramitochondrial acetyl-CoA located in the endoplasmic reticulum (ER) and nucleus. There are no acetyl-CoA-synthesising enzymes in the ER lumen. Therefore, acetyl-CoA has to be transported to the ER lumen by acetyl-CoA transporter 1(AT-1), a member of the multiple SLC33 transporters family, located in its membrane. This transport is facilitated by ACSS2 and ACLY, which are bound to ER membranes close to AT-1, thereby synthesising acetyl-CoA directly to the transporter site [177,197]; this assures the efficient transport and maintenance level of acetyl-CoA in the ER lumen required for acetylation of the proteins [198]. It is able to maintain an acetyl-CoA level sufficient for activation by transient acetylations of lysine groups of different proteins in the ER lumen, by specific lysine protein acetyltransferases 1 and 2 (ATase1,2, NAT8B) [199,200] (Figure 4). Expressions of AT1, ATase1 and ATase2 genes are regulated in an orderly manner by promoters that bind the same neuron-related transcription factors, CREB, c-FOS and c-JUN. On a post-transcriptional level, ATase 1 is acetylated, playing the role of an allosteric regulator of acetyl-CoA influx into the ER. ATase 2 is not acetylated but is primarily regulated on a transcriptional level by CREB, c-FOS and c-JUN, yielding activation of a secretory pathway. Neuronal ATases respond to environmental signals regulating the efficiency of the secretory pathway for molecular alterations linked, among others, with learning and memory formation [200,201]. Only properly folded proteins are acetylated and directed to the secretory pathway, whereas those with defective structure remain un-acetylated and are disposed by autophagy [200,201]. 

Such selective acetylation has been demonstrated, among others, for the β-amyloid precursor protein cleaving enzyme 1 (BACE 1), low-density lipoproteins receptor (LDLR) or amyloid precursor protein (APP), tubulin, and chaperons such as p53 or RE1-silencing [75,100,202,203]. The absolute concentration of acetyl-CoA in ER was not assessed. However, it may be assumed that it may be close to or somewhat higher than that in the cytoplasmic compartment [14,199]. The Km values for acetyl-CoA for AT-1 transporter in ER membranes are in the range of 0.010–0.014 mM, which are close to the cytoplasmic concentrations of this metabolite [14,199]. Therefore, the rate of acetyl-CoA influx into the ER compartment may change in accordance with increases and decreases in cytoplasmic levels of this metabolite that take place during neuronal maturation or excitotoxic injury, respectively [98,199,204]. Both in neuronal and glial cells, ACLY bound with ER provides acetyl-CoA for acetylation of reticular proteins through AT1 transporter and ATases 1/2 in the ER lumen [202]. The ER acetylating system functions in concert with the SLC25A1 mitochondrial malate/citrate antiporter and plasma membrane bond SLC13A5 sodium/citrate symporter (Figure 4). Overexpression of either gene in mice causes apparent over-acetylation of endoplasmic reticular proteins and the appearance of an autism-like phenotype with white matter disruption and altered neuron morphology [130,205]. On the other hand, heterozygous mice with point mutation of AT1S113R displayed decreased acetyl-CoA transport into ER, resulting in neurodegeneration and a propensity to infection, inflammation and cancer [206]. Mice homozygous for this mutation died at an early age due to fatal developmental arrest.

On the other hand, over-expression of AT1/SLC33A1in Tg mice caused a marked increase in acetylation lysine residues in brain ER proteins. This was accompanied by decreases in acetyl-CoA levels in the cytosol of the H4 neuroblastoma, hippocampus and isolated neurons, limitations in dendritic branching and development of an autistic phenotype [207]. These findings may be explained by the increased transport of acetyl-CoA into the ER, which depleted this metabolite from the cytoplasm, thereby affecting fatty acid synthesis and ACh synthesis, which are crucial for normal development of the brain [14,203,205]. Similar disturbances in brain development, metabolism and morphologic phenotype were reported in Tg mice with duplicated SLC13A5/sodium citrate or SLC25A1/CIC genes coding other citrate transporters, respectively [130,205]. These data indicate that the ACLY-AT-1 complex plays a critical role in the maintenance of an intra-ER acetyl-CoA level within safe limits.

In addition, the protein disconnected-interacting homolog 2A (DIP2A) is involved in the synthesis of acetyl-CoA and promotes activatory acetylation of cortactin [208] (Table 1). The latter participates in the development of dendritic spines and postsynaptic densities, regulates neuronal migration and is expressed abundantly in all brain regions containing pyramidal neurons [209]. Furthermore, knockdown of the DIP2 gene in mice resulted in impairment of neurotransmission, a decreased acetyl-CoA level in the cortex and presentation of autism-like repetitive behaviours and defects in social novelty. Application of acetylation cortactin mimetics ameliorated this pathology [209]. Conversely, histone deacetylase 6 (HADAC6), which decreases acetylation of cortactin, was found to promote proper neuronal migration of pyramidal neurons and dendrite development [210].

### 8.2. Nuclear Acetyl-CoA

The nuclear membrane, with its pores of 5–10 nm diameter, assures free direct access of cytoplasm-born acetyl-CoA to its target proteins, which should meet the demand for acetylations taking place in this compartment. Moreover, metabolites with much lower m.w., such as lactate/pyruvate, acetoacetate/BHB, acetate and acetylcarnitine, may move freely from cytoplasmic to intra-nuclear space. These substrates can also be direct donors of acetyl-CoA due to the presence of four enzymes of its synthesis in the nucleus, including ACLY, ACSS2, PDHC and carnitine acetyltransferase [16,211] (Figure 4). Thereby, nuclear acetylations may be secured by two pools of acetyl-CoA generated in intra- and extranuclear compartments, respectively. This would make nuclear acetylations partially independent of fluctuations in the acetyl-CoA level in the cytoplasmic neuronal compartment [14,98]. Nuclear acetyl-CoA supplies acetyl units for acetylations of several hundred proteins that include the most abundant histones, transcription factors, chaperons and enzymes, resulting in their post-translational modifications of activity, substrate affinity, resistance to degradation, etc. [16]. The degree of acetylation of specific histones on lysine residue is regulated reciprocally by the specific histone acetytransferases (HAT) and histone deacetylases (HADAC), respectively. As such, histones modulate the activity of several promoters regulating gene expression in an acetylation-dependent manner [16]. 

Multiple histone acetyltransferases and deacetylases were found to carry on regulatory acetylations/deacetylations of hundreds of histones and transcription factors that shape phenotype, plasticity and memory/cognitive functions of neuronal and glial brain cells [176,202]. In the nucleus, acetylations and deacetylations of chromatin histones are executed by multiple lysine histone acetyltransferases and histone deacetylases, respectively. The degree of histone acetylation plays a key role in the regulation of promoter sites and gene expression covering a vast range of phenotypic modifications on the levels of individual cells, tissues and whole organism, from fertilisation to death [16].

In the majority of physiologic and pathologic conditions, a higher level of acetyl-CoA in the cytoplasm was usually linked with better viability and functional parameters of brain cells, which may not always be the case [14,16]. For example, starvation of cultured human or mice cells lowered the acetyl-CoA level in the cytoplasm and decreased the activity of P300 acetyltransferase, yielding a drop in P300 acetylation and activation of autophagy [209]. On the contrary, cell feeding, or activation of the PDHC by DCA or lipoic acid, reconstituted the acetyl-CoA pool and decreased autophagy [212].

ACLY located in nuclei may provide acetyl-CoA directly to different histone lysine transacetylases (HAT), forming an ACLY-HAT-Histone functional complex. In cultured primary astrocytes, ACLY is accumulated in the nucleus in the complex with fatty acid binding protein 7 (FABP7), where it supplies acetyl-CoA directly to histone transacetylases, increasing the content of histone 3 acetylated on lysine 27 (H3K27ac) [213]. The levels of acetyl-CoA and ACLY in whole FABP7 KO astrocytes and isolated nuclei were, respectively, 25–30% lower than in WT controls. On the other hand, transfection with the nuclear localisation signal (NLS)-FABP7 gene caused the elevation of ACLY and a threefold increase in nuclear acetyl-CoA level and histone acetylations [213] (Table 1). This interaction modulates epigenetic modifications of several genes, including caveolin 1. It suggests that ACLY-derived acetyl-CoA may indirectly modify caveole-induced activity of astrocytes and proliferation of glioma cells [213,214]. Besides ACLY, ACSS2 may also provide acetyl-CoA for histone acetylations in the nuclei [16] (Figure 4). These two pathways appeared to be fully interchangeable, as acetyl-CoA concentrations in isolated nuclei were similar irrespective of the precursor citrate or acetate being used [215] (Table 1).

Differentiation of the catecholaminergic cell line (CAD) caused the translocation of ACSS2 into the nucleus, yielding increased acetylation of H3K9, H4K5 and H4K12 histones [176]. A similar interaction of ACSS2 with neuronal nuclei was demonstrated in vivo in the mouse hippocampus. Knockdown of ACSS2 in the hippocampus decreased histone acetylation and was accompanied by impaired object location memory and defective upregulation of immediate early genes following training. This indicates that ACSS2 associated with chromatin provides acetyl-CoA directly to histone acetyltransferases, which are acetylate histones that activate genes involved in neuronal differentiation, memory formation and different cognitive functions [16,177]. These data suggest that acetyl-CoA synthesised within nuclei by ACSS2 may be a key regulator of nuclear acetylations supporting the viability of neuronal cells. Hence, this enzyme constitutes a potential target for the development of therapies to treat neurodegenerative diseases. However, it remains unclear as to the fractional contribution of acetyl-CoA synthesised in the cytoplasm and acetyl-CoA synthesised in the nucleus in acetylations of nuclear proteins in the brain [175,177]. This is due to the fact that intra-nuclear synthesis of acetyl-CoA was found to be catalysed by ACSS2, ACL and PDHC [30,177,213] (Figure 4). Thus, the PDHC in cultured fibroblast was translocated from the mitochondria to the nucleus in stressful conditions during cell progression to generate nuclear acetyl-CoA indispensable for core histone acetylations supporting their entry to phase S [31,211]. Due to the large diameter of the PDHC (45 nm), it enters the nucleus through the non-canonical direct pathway from mitochondria tethered to the nuclear envelope by the NFT2 protein [210,216]. The significance of this pathway was confirmed by treatment of the cells with E1 subunit-specific siRNA, which depleted the nuclei of the E1 gene and decreased the concentration of acetyl-CoA and histone acetylation [31].

During four-cell porcine zygotic genome activation, the PDH E1 subunit was translocated to the nucleus, increasing E1 and mRNA E1 levels and producing acetyl-CoA, which increased acetylation of histones H3K27ac and H3K9a [217]. Overexpression of E1 increased levels of acetylated histones, whereas its targeting reduced histone acetylation and inhibited zygotic genome activation. These data indicate that E1 translocation between the mitochondria and nucleus plays a key role in the first stages of embryonal development [217]. It is worth emphasising that each of the reports claims the prevalent role of the tested pathway in the generation of nuclear acetyl-CoA [30,176,213]. This controversy could be elucidated by a study that would investigate all acetyl-CoA-generating pathways in one experiment. In addition, the presence of a nuclear pool of carnitine acetyltransferase was detected in peripheral tissues and non-neuronal cell lines [218]. No study was performed to determine its presence in neuronal cell nuclei in the brain. Irrespective of these uncertainties, concentration-dependent regulatory properties of the nuclear pool of acetyl-CoA in the acetylation of nuclear proteins remain in concert with our hypothesis concerning the neuroprotective effects of high levels of acetyl-CoA in the mitochondria [14,15,101,131]. This thesis is supported by the antidepressant effects of acetate supplementation in depressive mice, which were accompanied by increased levels of acetyl-CoA and histone acetylations in their brains [178] (Table 1). Thus, mitochondrial acetyl-CoA may be a key metabolite in protecting neurons against different neurodegenerative signals, whereas cytoplasmic acetyl-CoA in cholinergic neurons would regulate rates of acetylcholine synthesis and cholinergic neurotransmission responsible for the maintenance of cognitive functions, as well as intranuclear acetylations [14,15,175,177]. However, nuclear histone acetyltransferases Tip 60, 8KAT8 (histone 4) and p300/CPB show high affinities to acetyl-CoA, with Km values being in the range of 0.0003–0.0048 mmol/L [219,220,221]. Therefore, intraneuronal acetyl-CoA levels being in the range 0.008–0.015 mmol/L may saturate these enzymes, assuring submaximal rates of in situ nuclear histone acetylations. As a result, changes in nuclear acetyl-CoA levels may exert a relatively low impact on rates and yield of HATs reactions [14].

This raises the hypothesis that alterations in the ratio expression of HAT to histone deacetylases may be a more important regulatory factor in histone acetylation [177]. Ethanol application (5 g/kg) to 21-day-old rats, simulating adolescent binge exposure, caused a loss of basal forebrain cholinergic neurons, measured as decreases in the level of ChAT, p75NTR NGF receptors and vesicular ACh transporter, along with fast excessive methylation and acetylation of H3K9 [222]. These negative effects of ethanol were reversed by simultaneous application of galantamine, an acetylcholinesterase inhibitor, which decreased methylation of H3K9. Both ethanol and high doses of acetate induced H3K9 and H3K27 histone acetylation, indicating that ACSS2 takes part in this phenomenon by supplying acetyl-CoA for HATs contributing to epigenetic alcohol-linked pathology [177]. However, it is not clear how prolonged drinking links genetic alterations with addictive behaviour [223].

## 9. Conclusions

Glucose or lactate-derived pyruvate through PDHC reaction is a principal source of acetyl-CoA for energy production in the TCA cycle in the mitochondrial compartment of all brain cells. Other energy substrates, such as BHB and acetate, may only in part and in specific pathophysiological conditions replace pyruvate as direct precursors of acetyl-CoA (Figure 1). A small fraction of acetyl-CoA is utilised in a vast number of acetylation reactions, taking place in all subcellular compartments of brain cells. Under resting conditions, rates of oxidative metabolism in the brain are about 10 times faster than in peripheral tissues. In addition, metabolic profiles of acetyl-CoA may differ markedly in individual neuronal and glial cell groups. In particular, the rate of energy metabolism in neurons is several times higher than in glial cells. This makes the maintenance of equilibrium between synthesis and utilisation of acetyl-CoA a critical point for the proper functioning and viability of neuronal cells. Therefore, knowledge about the cellular and intracellular compartmentalisation of acetyl-CoA is necessary for an explanation of its diverse roles in healthy and diseased brains (Figure 1, Figure 2, Figure 3 and Figure 4).

As such, concentrations of acetyl-CoA in different subcellular compartments are low and may change in a relatively broad range in different physiologic and pathologic conditions. Therefore, alterations in acetyl-CoA concentration themselves may be early signals affecting cell phenotype and genotype. The differentiation of septal cholinergic neurons caused a shift of acetyl-CoA from the mitochondrial to cytoplasmic compartments, stimulating ACh synthesis but increasing their susceptibility to neurotoxic signals. There is a direct correlation between cytoplasmic acetyl-CoA level and content and rate of ACh synthesis in cholinergic neurons (Figure 2). The rate of NAA synthesis by NAT correlated positively with the acetyl-CoA level in neuronal mitochondria and neuron viability. TBI post-impact infusion of glucose or BHB increased acetyl-CoA in the peri-contusional brain area and improved outcome from the injury. 

Multiple enzymes in the nucleus—ACLY, ACSS2, PDHC—may interchangeably supply sufficient amounts of acetyl-CoA for histone lysine residue acetylations by HATs. The level of histone acetylation correlated with the content of acetyl-CoA in the nucleus. Long-term feeding with ketone esters or glycerol tri-acetate increased acetyl-CoA levels in the brains of Tg AD mice and prevented their cognitive impairment. Overexpression of the AT1 transporter or SLC25A1 carrier on the ER membrane caused an increase in the intra-ER acetyl-CoA level and aberrant protein acetylations that impaired the secretory pathway (Figure 4). These data demonstrate that the identification of metabolic pools of acetyl-CoA in different neuronal and glial compartments is of principal significance for target-oriented approaches in the management of neurodegenerative diseases. The cellular specificity of such a procedure would also be important for the reduction in apparent side effects of acetyl-CoA-modifying compounds linked with its central role in overall metabolism.

**Table 1 ijms-23-10073-t001:** Levels of acetyl-CoA in different brain compartments in various experimental models of brain pathologies. Historic perspective.

Experimental Model	Signal/Conditions	Acetyl-CoA Level/Relative Change	Reference/Comments
Rat brain	Hypoxia in vivo	Whole tissue *(nmol/g tissue)*	[17]
Control	5.4
Hypoxia 100N_2_ 90 s	6.7 **
Rat brain	Brain region (whole tissue)	Whole tissue *(nmol/g tissue)*	[224]
Thalamus	9.1
Hippocampus	7.1
Cortex	6.2
Cerebellum	6.1
Rat brain slices	60 min. incubation 31.2 mM K^+^	Brain slices *(nmol/g tissue)*	[18]
Control	5.04
+3-bromorypuvate 0.25 mM	2.45
Rat brain synaptosomes	30 min. incubation 30 mM K^+^	SynaptosomesMitochondria Cytoplasm	[19]

(*pmol/mg protein*)
Control	12.3 46.8
+3-bromopyruvate 0.25 mM	0 7.4 **
Healthy adult rat brain synaptosomes	Healthy control	Whole synaptosomes (*pmol/mg protein*)	[143] different from pyruvate alone, ^†^ *p* < 0.05
Substrate used (mM)
Pyr. 2.5 BHB 2.5 Pyr. + BHB
24.3 7.1 22.8
STZ diabetes 10 d	31.3 * 10.5 * 29.4 *
Streptozotocin-diabetic rat brain synaptosomes	STZ diabetes 10 d + Insulin 5 d	30.6 * 10.0 *^†^ 35.6 *^†^
Cholinergic neuroblastoma cell culture: nondifferentiated (NC) and differentiated (DC, db-cAMP 1 mM + retinoic acid (RA) 0.001 mM 48 h)	Control	Cellular compartment levels*(pmol/mg protein)*	[98]from respective NC, ^†^ *p* < 0.05, ^††^ *p* < 0.01
Mitochondria Cytoplasm
NC DC NC DC
71 22 ^†^ 13 50 ^†^
+NGF 100 ng/mL 24 h	55 42 ** 71 ** 29 *^†^
Native SN56TrkA-/p75^NTR^+	Control	95 23 ^†^ 13 49 ^†^
Tg T17 SN56TrkA+/p75^NTR^+	+NGF 100 ng/mL 24 h	59 * 39 * 129 ** 48 ^†^
Cholinergic neuroblastoma cell cultureTg T17 SN56TrkA+/p75^NTR^+ NC, and DC	24 h cell culture with:	Relative change against no addition control *(%)*	[97,126]Different from respective NC, ^†^ *p* < 0.05, ^††^ *p* < 0.01; from Aβ (25–35) alone, ^‡^ *p* < 0.05, ^‡‡^ *p* < 0.01

Mitochondria Cytoplasm
NC DC NC DC
Aβ25-35 0.001 mM	10 −23 −17 −58 **
Acetyl-carnitine 0.1 mM	+39 ** 0 ^†^ 0 +54 **^††^
Aβ + acetyl-carnitine	+22 ^‡‡^ 0 0 0 ^‡‡^
ILβ 10 ng/mL	−11 −18 +38 * −42 *^††^
	Aβ + ILβ	−18 −1 +1 +3 ^‡‡^
Cholinergic neuroblastoma cell culture	ChAT (*nmol/min/mg protein*)NC DC	Whole cells (*pmol/mg protein*)NC DC	[100]Different from respective native SN56, ^†^ *p* < 0.05, ^††^ *p* < 0.01
Native SN56 TrkA-/p75^NTR^+	0.22 0.79 ***	31.2 21.9 ***
Tg T17 TrkA+/p75^NTR^+	0.19 0.47 ***	39.7 ^†^ 26.8 ***^†^
Tg ChAT2 TrkA-/p75^NTR^+	3.80 ^†††^ 6.80 ***^†††^	15.5 ^††^ 11.2 *** ^††^
Cholinergic neuroblastoma cellsNative SN56 TrkA-/p75^NTR^ + DC	24 h cell culture with:Control	Mitochondria Cytoplasm(*pmol/mg protein*)	[40]^†^ different from ZnCl_2_ 0.10 mmol/L
11.8 20.9
ZnCl_2_ 0.10 mM	9.3 19.6
ZnCl_2_ 0.15 mM	11.4 13.5 *^†^
Cholinergic neuroblastoma cellsNative SN56 TrkA-/p75^NTR^ + NC and DC	30 min incubation (protein free medium) with:Zn 0.1 mM	Relative change vs. no Zn control *(%)*Mitochondria CytoplasmNC DC NC DC−5 −35 ** −100 ** −80 **	[110]
Subcutaneous pyrithiamine (PT) 0.025 mg/kg b.w./day and thiamine free diet 14 dRat forebrain synaptosomes	PT synaptosomes vs. no PT control	Forebrain synaptosomesRelative change against no PT control *(%)*	[119]

Mitochondria Cytoplasm
No Ca Ca 1.0mM no Ca Ca1.0mM
−53 *** −35 *** −43 *** −24 *
Subcutaneous PT 0.025 mg/kg b.w./day and thiamine-free diet 14 d. Rat forebrain whole mitochondria	PT whole forebrain mitochondria vs. no PT control	Forebrain whole mitochondriaRelative change vs. no PT control *(%)*	[120]
No Ca Ca 0.01 mM ADP/HX
−62 *** −62 *** −52 ***
Cholinergic neuroblastoma cell cultureNative SN56 TrkA-/p75^NTR^+	Thiamine-free culture medium 48 h+Amprolium 2 mM	Relative change vs. no amprolium NCcontrol *(%)*	[121]Amprolium suppressed TPP level—28% vs. control
Mitochondria Cytoplasm.
NC DC NC DC
−43 −57 −58 *** −50 **
Endoplasmic reticulum from WT and AT 1-^1S113R/+^ mice	Mutation *AT 1-^1S113R/+^*	Acetyl-CoA *transport (pmol/mg/5 min.)*	[206]
WT 370
AT-1^S113R/+^ 142 ***
N9 microglioma cells culture	24 h culture with:LPS 0.01 µg/mL	Relative change against respective no addition control *(%)*	[38]^‡‡^ different from SNP 0.4 mM, *p* < 0.01^†††^ different from N9 cells, *p* < 0.001
Whole cells
N9 SN56
−23 * +4
SynchronizedCholinergic neuroblastoma cellsNative SN56 TrkA-/p75^NTR^+ DC	SNP 0.4 mM	−3 −38 *
LPS + SNP	−6 92 ***^†††‡‡^
WT 14–16 mos mouse brain AβPP-Tg 2576 14-16 m mouse brain	Accumulation about 0.6 μM Aβ_1-42_ in Tg brain	Relative change vs. WT control *(%)*Mitochondria Cytoplasm**	[62]Acetyl-CoA—control WT miceSynapt. mitoch. 39 *pmol/mg prot.*Synapt. cytopl. 90 *pmol/mg prot.*Whole brain mitoch. 45 *pmol/mg.*
Forebrain synaptosomes	−44 ** −34
Forebrain whole mitochondria	+5 -
WT mouse brainAT1 Tg mouse brain (overexpression)	Hippocampus Isolated adult neurons H4 neuroglioma	AT1 Tg vs. WTRelative difference *(%)*Cytoplasm	[207]
−41 *
−45 *
−43 *
WT 9 d postnatal mouse brain	24 h post hypoxia/ischemia	Relative change vs. control *(%)*Mitochondrial fractionVehicle-treated DCA-treated+6 +27 *	[34]
Cell line culturesWT SN56 TrkA-/p75^NTR^ NC	Intracellular Zn accumulation of 5 nmol/mg protein at extracellular Zn in culture medium:0.125 mM	Relative change vs. no Zn control *(%)*SN56 NC −54 ***	[11]^†^ different from NC, *p* < 0.05
DC	0.110 mM	SN56 DC −48 ***^†^
SHSY5Y dopaminergic neurons	0.150 mM	SHSY5Y −31 *
C6 astroglioma	0.200 mM	C6 −44 **
3XTg AD 16.5 mos mouse brain	8 mos ketone ester-feeding	Relative change vs. non-ketotic control*(%)*Hippocampus +79 *	[134]Acetyl-CoA no ketone control: 17 μmol/g tissue
Mouse BV2 microglial cells culture	Dimethylsulfoxide-induced 6 h hypoxia	Relative change vs. no hypoxia control*(%)*+79 **	[30]
Hypoxia + Lonidamine 0.05 mM	−58 *
Hypoxia + 3-Bromopuryvate	−42 *
Cholinergic neuroblastoma cellsWT SN56 TrkA-/p75^NTR^+ DC	30 min incubation (protein-free medium) with:ControlNifedipine 0.01 mMGVIA 0.0005 mMMVIIC 0.0002 mM	Whole cells (*pmol/mg protein*)No Zn Zn 0.15 mM30.5 13.8 *30.7 29.2 ^†^28.8 21.6 *^†^28.1 20.5 *^†^	[105]Compounds used here are inhibitors of different types of calcium channels.* *p* < 0.01 vs. no Zn control; ^†^ < 0.01 vs. 0.15 mM Zn.
SAMP8 mice brain cortex	13 mos vs. 9 mos changeNo treatmentFed with CMS121 4 mosFed with J147 4 mos	Relative change 13 mos vs. 9 mos	[175]CMS121, J147 are acetyl-CoA carboxylase inhibitors.
*(%)*
−41 ****
−12
−6
HT22 hippocampal neuronal cell culturePrimary E21 mice neuronal culture	24 h culture with:	Relative change vs. no addition control*(%)*	[10,175]Compounds used here inhibit acetyl-CoA carboxylase by different mechanisms.
+ACC1 siRNA	+114 ***
+TOFA 0.01 mM	+178 ***
+CMS 121 0.001 mM	+140 ***
+J147 0.001 mM	+100 **
+CAD031 0.001 mM	+177 ***

+CMS 121 0.001 mM	+57 ***
+J147 0.001 mM	+29
+CAD031 0.001 mM	+108 ***
Brain-specific *pdha1^flox8/wt^* deficient mice (PDHD)	PDHD	Relative change vs. control *(%)*−12	[54]
3xTgAD miceWT control mice	Ageing—2, 11, 21 mos hippocampus whole tissueControl	2 mos 11 mos 21 mos	[179]Different from the corresponding 2 mos mice, ^†^ *p* < 0.05, ^†††^ *p* < 0.001
*(Arbitrary units)*
Male
0.5 1.1^†^ 1.3 ^†††^
3XTgAD	1.2 * 1.6 * 2.6 **^†††^
	Female
Control	0.5 0.8 1.0 ^†^
3XTgAD	0.5 1.3 *^†^ 1.2 ^†^
Rat permanent middle cerebral artery occlusion model of brain stroke (pMCAO)	Shengui Shanseng San (SSS) extraction feeding per os 3 d before and 7 d after pMCAO	Relative change vs. sham control	[35]Absolute sham control value of infarct-corresponding control region equal to 24.4 µmol/µL tissue is 10^6^ times higher than those reported elsewhere.
In brain infarcted region *(%)*pMCAO −80 ***
Low dose SSS + pMCAO −52 ***
Middle dose SSS +pMCAO −44 ***
High dose SSS +pMCAO −4
Closed-head impact acceleration model of mild or severe traumatic rat brain injury (mTBI/sTBI)	mTBI/sTBI	Relative change vs. control *(%)*Whole brain extracts	[58]Absolute control value about 39 nmol/g wet weight is about 10 times higher than values reported elsewhere.Different from the corresponding of post mTBI time, ^†^ *p* < 0.005
Post mTBI 6 h −13
24 h −22
48 h −24
120 h −13
Post sTBI 6 h −34 *
24 h −56 *^†^
48 h −47 *^†^
120 h −58 *^†^
HEK293 cell culture	DIP2A overexpression	Relative change DIP2A vs. no insert control *(%)*+120 *	[208]
Traumatic brain injury/control cortical impact rat brain (TBI/CCI)	TBI/CCI	Peri-contusional brain cortex acetyl-CoA*(ng/mg protein)*Early immediate 3 h i.v. administration	[41]Absolute control value is about 34.5 pmol/mg protein.
Sham (saline 0.9%) 27
Control 38
Glucose 30% 57 *
Lactate 100 mM 29
BHB 2M 52 *
Late (6 h post impact) 3 h i.v. administration
Glucose 30% 38
Lactate 100 mM 21
BHB 2M 38
Cholinergic neuroblastoma cellsWT SN56 TrkA-/p75^NTR^+ DC	30 min incubation (protein-free medium) with:	Relative change vs. control *(%)*Mitochondria Cytoplasm	[160]Mecamylamine is a nonselective antagonist of nicotinic receptors. 2APB is inhibitor of IP3 receptors and TRP channels.
Mecamylamine 0.002 mM	−36 ** +7
Nifedipine 0.01 mM	0 +28
2-Aminoethoxydiphenyl borate (2-APB) 0.05 mM	+43 -56 **
Zn 0.15 mM	−64 *** −39 **
Human fibroblastoma HT1080 cell line ACLY WTACLY-WT ACLY KO	4 h incubation with or without 20 mM acetateACLY-WTACLY-KO	Relative change vs. WT-acetate control (*%*)	[215]Absolute control value for ACLY-WT is 6.1 μM (normalised to internal standard)
acetate 20 mM No acetate
0 −14
−67 *** −95 ***
E18 C57BL/6J mice model of AD	24 h culture withAβ_1-42_ 10µM	Relative change vs. control *(%)*	[183]Absolute control values for neurons and microglia are 0.45 and 0.75 μM, respectively
Neurons Microglia **
0 −31
5XFAD 9 mos mouse brain	5XFAD control 5XFAD + efavirenz 0.1 mg/kg b.w./d in drinking water from 3 to 9 mos of life	Whole brain Mitochondria(*pmols/mg protein*)	[225]Efavirenz is an inhibitor of reverse transcriptase.Acetyl-CoA control levels reported here are about 10 times higher than reported elsewhere.
145 87
351 *** 352***
B6SJ/L 9 months mouse brain	B6SJ/L control	361 157
Tg Cyp46a1^+/+^	Tg Cyp46a1^+/+^	257 *** 128
Tg Cyp46a1^−/−^	Tg Cyp46a1^−/−^	143 *** 100 ***
Cholinergic neuroblastoma cellsWT SN56 TrkA-/p75^NTR^+ NC and DC	24 h culture in thiamine-free medium with: +Zn 0.1 mM+Amprolium 5 mmol/L+Zn +Amprolium	Relative change vs. no Zn, and amprolium control *(%)*Mitochondria	[45]Absolute control acetyl-CoA levels in NC and DC mitochondria were: 11.6 and 11.9 pmol/mg protein, respectively. Absolute control acetyl-CoA levels in NC and DC cytoplasm were: 13.6 and 11.7 pmol/mg protein, respectively.^††^ different from NC/DC Zn, *p* < 0.0.1different from NC/DC amprolium, ^‡^ *p* < 0.05, ^‡‡^ *p* < 0.01
NC DC
−5 −23 **
−5 −16 *
−45 ***^††‡‡^ -50 **^††‡‡^
Cytoplasm
+Zn 0.1 mM	−4 −12
+Amprolium 5 mmol/L	−17 −12
Thiamine-deficient culture medium	+Zn +Amprolium	−54 **^††‡‡^ −53 **^††‡‡^
C6 astroglioma cellsCholinergic neuroblastoma cellsWT SN56 TrkA-/p75^NTR^+ DC	24 h culture C6 in thiamine-free or thiamine-supplemented medium with:Amprolium 10 mMZn 0.15 mMZn 0.20 mM24 h culture SN56 in thiamine-free medium in co-culture with C6C6 co-culture Amprolium 5 mMZn 0.1 mM Amprolium + Zn Amprolium + Zn+C6 co-culture	Relative change vs. no Zn, no amproliumcontrol *(%)*Thiamine deficient Thiamine suppl.	[39]Absolute control levels of acetyl-CoA in SN56 and C6 cells were:27.2 and 14.6 pmol/mg protein, respectively.^†^ different from Amprolium+Zn, *p* < 0.05
−26 ** 0
−28 −16
−68 ** −56 **
Relative change vs. no co-culture, Zn, and no amprolium control (*%*)
+10
−26
−29
−64 *
−10 ^†^
WT mouse brainC57BL/6J mouse brain	Glycerol triacetate 3 g/kg b.w./d 10 d by gavage, and euthanised 60 min. post last gavage	HippocampusRelative change vs. control *(%)*	[178]
Whole tissue Nuclei Cytoplasm

+171 * +19 *** +13 **
Non-fasted mouse brain	Sacrificed 30 min. post oral ketone esters (KE) administration 3 mg/g b.w	Relative change KE vs. control *(%)*Brain cortex+114 ***	[226]
Cultured primary neurons (E17 C57BL/6J mice)	Astrocyte-derived ApoE particlesAstrocyte-derived medium (ADM)Apo E enriched ADMApo E depleted ADM	Relative change vs. no ApoE control*(%)*Acetyl-CoA/CoA ratioWhole cells Nuclei+86 * +175 ***Acetyl-CoA/CoA ratio+200 ***+40	[32]
WT mouse brainElp3 conditional KO mouse brain	Lack Elongator to Atat1 activity	Relative change vs. WT control *(%)*Cortical neurons−72	[75]
WT mouse brainC57Bl/6J mouse brain	Acute stress	Relative change vs.no stress control*(%)*Prefrontal cortex+113 *	[192]Absolute acetyl-CoA level, 0.37 pmol/μg
C57BL/6J mice—stroke and hypoxia	12 wk post-stroke oral administration p75 ^NTR^ modulator (LM11A-31)	Relative change vs. sham control *(%)*Brain infarcted regionNone LM11A-31−32 +36 *	[23]
Primary astrocytes—0–1-day-old mice cerebral cortexU87 human glioblastoma cellsU87FABP7wtU87FABPmut.U251human glioma cells U251 FABP7KO	FABP7-KO vs. WT cellsFABP7wt vs. controlFABP7mut vs. controlFABP7KO vs. control	Relative change vs. WT control *(%)*Whole cells Isolated nuclei−34 * −28 *+87 * +74 *−10 −39−48 * −70 *	[213,214]Absolute acetyl-CoA for control WT cells is 450 pmol/10^6^, and 74 pmol/10^7^ nuclei.
WT mouse brainSLC25A1 nTg mouse brain	Hippocampus and cortex cytoplasmLumen of the endoplasmic reticulum	Relative change vs. WT control *(%)*Cytoplasm ER+58 *** +72 ****	[130]SLC25A1 nTg—mitochondrial citrate carrier

The majority of the data are presented as relative (%) change versus respective control value. This results from the fact that they are presented in arbitrary units. In some cases, absolute values of acetyl-CoA are given to enable quantitative assessment of this metabolite. Distribution data are deleted for clarity. Significance of differences between groups is marked by superscript symbols. Data significantly different from respective control, * *p* < 0.05; ** *p* < 0.01; *** *p* < 0.001; **** *p* < 0.0005. Other comparisons are given as individual references. Abbreviations: ACLY, ATP citrate lyase; AD, Alzheimer’s disease; ADM, astrocyte-derived medium; Atat1, α-tubulin N-acetyltransferase 1; CCI, controlled cortical impact; DCA, dichloroacetate; DH, stroke distal middle cerebral artery occlusion with hypoxia; Elp3, cKO mouse with conditional loss of Elp3 in cortical progenitors; EFV, efavirenz CYP46A1 activator; ER, endoplasmic reticulum; FABP7, fatty acid binding protein 7; GTA, glyceryl triacetate; mTBI/sTBI, mild/severe traumatic brain injury; p75NTR, p75 neurotropin receptor; PDHD, pyruvate dehydrogenase deficiency; pMCAO, permanent middle cerebral artery occlusion; PT, pyrithiamine; SSS, Shenggui Sansheng San composed of Panax ginseng root and rhizome, Angelica sinensis root and rhizome, Cinnamomum cassia; TBI, traumatic brain injury.

## Figures and Tables

**Figure 1 ijms-23-10073-f001:**
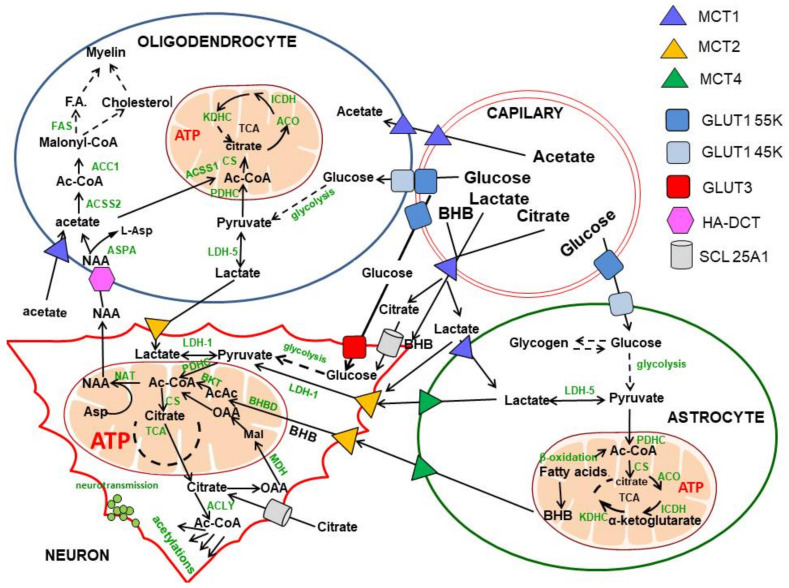
Intracellular and intercellular compartmentation of acetyl-CoA and precursors in the brain. The principal brain energy substrate, glucose, crosses the BBB though the Glut55kD transporter and enters oligo, micro and astroglial cells through Glut45kD. Neurons take up glucose by the Glut3 high-affinity transporter, consuming 60–80% of its overall brain pool. The final product of glycolysis—pyruvate through mitochondrial PDHC reaction—is the principal source of acetyl-CoA for energy production in neurons and glial brain cells. Neurons may also utilise lactate, BHB or citrate provided either for circulation and/or by astrocytes and oligodendrocytes; these two classes of glial cells produce a surplus of lactate due to relatively low rates of oxidative metabolism. Neuronal mitochondria are the only compartment synthesising NAA, which is transported to oligodendrocytes serving as a main source of acetyl units for fatty acid and cholesterol synthesis for myelin formation. Astrocytes are brain cells with a high rate of fatty acid oxidation net producing BHB. Moreover, in ketonemic conditions, BHB crosses the BBB by MCT1 in a concentration-dependent manner. Neurons take up BHB by MCT2, converting it to acetyl-CoA. As a result, BHB may serve as an alternative energy substrate in hypoglycaemic conditions. At least five functionally distinct subcellular compartments in the neurons can be distinguished. They include mitochondria, where acetyl-CoA is utilised mainly in the TCA cycle coupled with the respiratory chain and ATP as well as with NAA synthesis. In the cytoplasmic compartment, acetyl-CoA is produced from citrate, acetyl-carnitine acetoacetate or acetate in respective enzymatic reactions and used for fatty acid and cholesterol synthesis or protein, peptide or lipid acetylations. Separate pools of acetyl-CoA are formed by the nucleus and endoplasmic reticulum, where histone and protein acetylation takes part in the regulation of gene expression, protein functionalisation and disposal. Cholinergic neurons constitute a specific neuronal sub-compartment in which ACh synthesis and the cholinergic neurotransmission process takes place (see Figure 2). The neuronal axonal compartment contains unique tubulin-bound protein acetylation, which is responsible for axonal transport (see Figure 3). This figure demonstrates that indirect transport of acetyl units between different cell types and internal subcellular compartments plays a crucial role in the maintenance of brain homeostasis. Abbreviations: Enzyme names in the boxes: ACC1, acetyl-CoA carboxylase 1; ACCS1, acetyl-CoA synthase; ACO, aconitase; ACLY, ATP-citrate lyase; BKT, β-ketothiolase; CS, citrate synthase; GDH, glutamate dehydrogenase; HBDH, β-hydroxybutyrate dehydrogenase; HMGR, β-hydroxy-β-methylglutaryl-CoA reductase; ICDH, isocitrate dehydrogenase; KDHC, ketoglutarate dehydrogenase complex; LDH, lactic dehydrogenase; NAA, N-acetyl-aspartate; NAT, aspartate N-acetyltransferase; PDHC, pyruvate dehydrogenase complex; Other abbreviations: AcAc, acetoacetate; BHB, β-hydroxy-butyrate; F.A., fatty acids; GLUT, glucose transporters; HA-DCT, high-affinity dicarboxylate transporter; MCT, monocarboxylate transporters; SLC25A1, mitochondrial carrier family.

**Figure 2 ijms-23-10073-f002:**
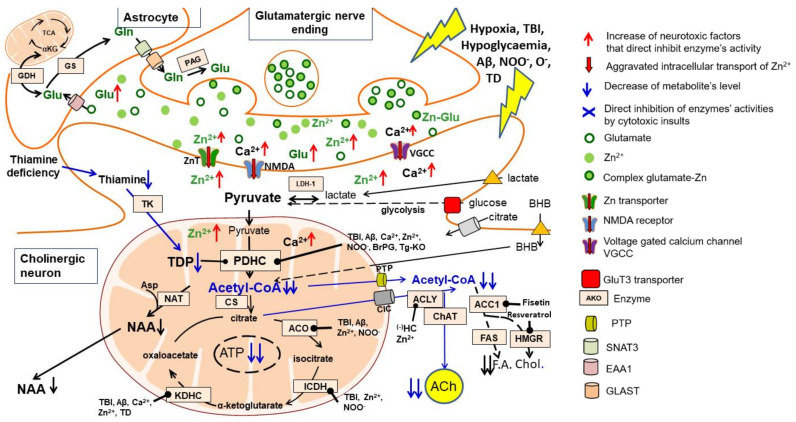
Excitotoxity-linked mechanisms of acetyl-CoA deficits in cholinergic neuron injury. Pyruvate is the final product of the glycolytic pathway and also the main precursor of acetyl-CoA generated in neuronal mitochondria by a PDHC-catalysed reaction. Pathologic signals associated with AD and other cholinergic encephalopathies, such as excessive synthesis/accumulation of Aß, episodes of hypoxia/hypoglycaemia and TBI, lead to prolonged depolarisation and excitotoxic stimulation of glutaminergic terminals releasing an excess of glutamate and Zn. This causes increased accumulation of zinc and calcium ions in postsynaptic neurons, stimulating the synthesis of oxygen and nitrosyl free radicals. All these cytotoxic signals directly inhibit PDHC activity. As a result, the synthesis and utilisation of acetyl-CoA in the TCA cycle is reduced, resulting in inhibition of ATP and NAA synthesis. There is also a reduction in acetyl-CoA transport out of mitochondria and the inhibition of hundreds of transacetylation reactions in the neuronal cytoplasm, endoplasmic reticulum and nucleus (see Figure 4). In cholinergic neurons, a significant fraction of the cytoplasmic acetyl-CoA pool is utilised for the synthesis of the neurotransmitter ACh. This process is facilitated due to the formation of an ACLY-ChAT complex. However, this additional need for acetyl residues makes cholinergic neurons more sensitive to neurotoxic signals than neurons of other neurotransmitter systems. TDP deficits increase the permeability of neuronal plasma membranes for Zn, Ca and other cytotoxic compounds leading to aggravation of their inhibitory effects on the PDHC and other enzymes of acetyl-CoA metabolism. The supply of alternative precursors of acetyl-CoA, such as lactate, BHB or acetate, may provide additional amounts of this metabolite, bypassing the PDHC and glycolytic pathway. Thereby, they may in part overcome deficit of pyruvate-derived acetyl-CoA. Moreover, inhibition of acetyl-CoA utilisation for lipid synthesis may preserve its pool for key energy-producing pathways. Abbreviations: Enzyme names in the boxes: ACC1, acetyl-CoA carboxylase 1; ACO, aconitase; ACLY, ATP-citrate lyase; ChAT, choline acetyltransferase; CS, citrate synthase; FAS, fatty acid synthetase; GDH, glutamate dehydrogenase; GS, glutamine synthetase; HMGR, β-hydroxy-β-methylglutaryl-CoA reductase; ICDH, isocitrate dehydrogenase-NADP; KDHC, ketoglutarate dehydrogenase complex; LDH, lactic dehydrogenase; NAT, aspartate N-acetyltransferase; PAG, phosphate activated glutaminase; PDHC, pyruvate dehydrogenase complex; TK, thiamine kinase; Other abbreviations: ACh, acetylcholine; Chol., cholesterol; F.A., fatty acids; NAA, N-acetyl-aspartate; SNAT, sodium-coupled neutral amino acid transporter; EAA, excitatory amino acid transporter; GLAST, glutamate and aspartate transporter. Blue arrows, decrease; red arrows, increase; black arrows with dot, inhibition.

**Figure 3 ijms-23-10073-f003:**
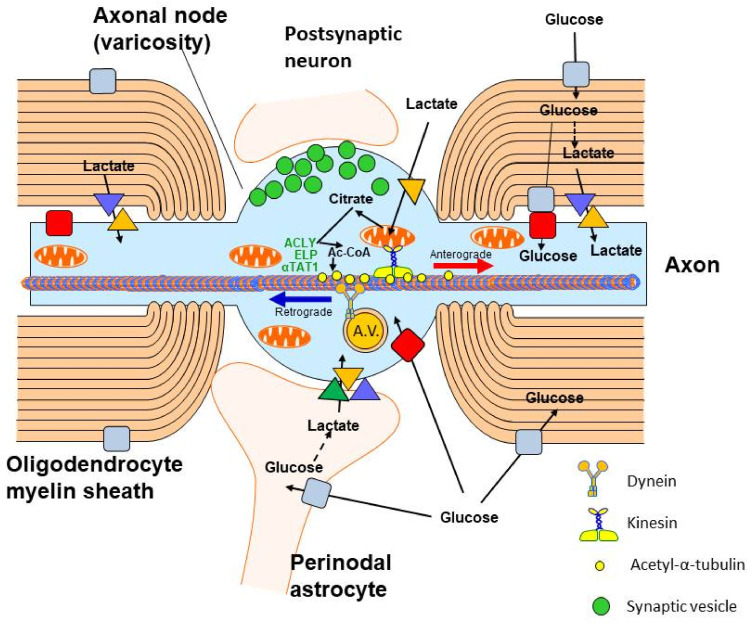
Acetyl-CoA metabolism in axonal neuron compartment. Glucose is transported to the axons through GLUT3. The lactate is taken up by MCT2 from extracellular space and from astrocytes adjacent to axonal nodes or from oligodendrocyte myelin sheaths through MCT1/MCT4. Acetyl-CoA generated in axonal mitochondria is transported to axonal cytoplasm by ATP-citrate lyase (ACLY), which forms the complex with elongator protein (ELP) and α-tubulin acetyltransferase 1 (αTAT1), which acetylates α-subunit of tubulin. Acetylation increases the rate of anterograde neurotubular transport of mitochondria, proteins and other compounds by the kinesin complex to support the function of nerve terminals. On the other hand, metabolic wastes are transported retrogradely by dynein within axonal vesicles (A.V.). Axonal nodes are also neurotransmission sites forming synapse with the dendritic spines of other neurons.

**Figure 4 ijms-23-10073-f004:**
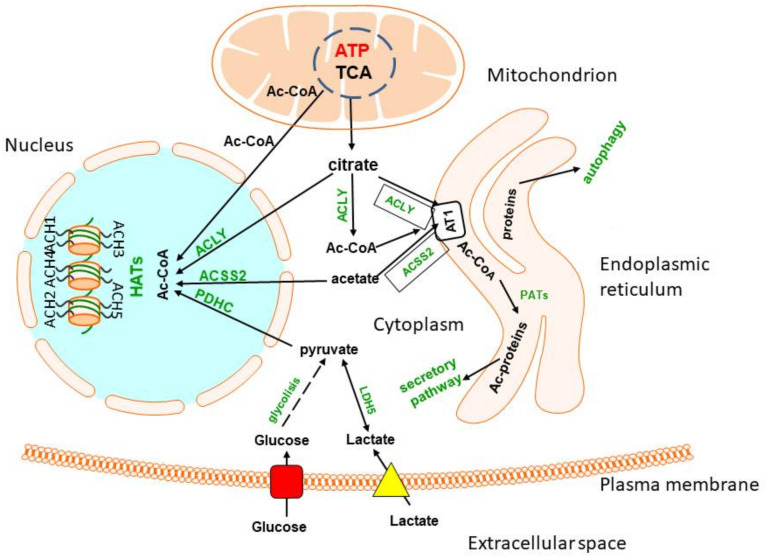
Acetyl-CoA metabolism in endoplasmic reticulum and nucleus. Acetyl-CoA is transported from the cytoplasm to endoplasmic reticulum by the acetyl-CoA transporter 1. It may be taken up directly from cytoplasm or generated by ACLY or ACSS2, forming functional complexes with AT1. Structurally competent proteins are then acetylated in ER lumen by protein acetyltransferases and released by a secretory pathway. Misfolded and damaged proteins are directed to autophagosomes. The nuclear membrane is fully permeable to cytoplasmic acetyl-CoA. However, some amounts of this metabolite are synthesised within the nucleus by ACLY, ACSS2 or PDHC from respective precursor metabolites. They provide acetyl-CoA directly to HATs, which acetylates multiple histones, thereby regulating expression of different parts of the genome.

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
