# Peer review of "Metabolic and Cellular Compartments of Acetyl-CoA in the Healthy and Diseased Brain"

_ijms, 2022, doi:10.3390/ijms231710073_

Round 1

Reviewer 1 Report

The English made the reading difficult.

There are some concepts that could merit a figure. For example lines 301-305:

Axonal 301 transport is a process that delivers organelles, proteins and low molecular weight species, 302 occluded in axonal vesicles, along microtubular tracks to distant neuronal axonal com-303 partments. They also convey retrograde transport. Neurotubuli consist of globular α and 304 β tubulin heterodimer subunits forming helical chains extending along entire axons. 305

There are no citations for this very important concept of transport of organelles between cells. As I mentioned before, a diagram or image will be very helpful.  

Similarly, lines 348-350 can also be included in the same diagram

anterograde and retrograde axonal transport is executed by motor-proteins 348 kinesin and dynein coupled with elongator complex, which supports α-tubulin acetyla-349 tion catalyzed by αTAT1located in axonal vesicles.

Author Response

Answer to Referee 1.

New citations are provided for Axonal acetyl-CoA chapter. We inserted a diagram – Fig.3  demonstrating precursors and acetyl-CoA metabolism in the axonal compartment. It should help in following the text.

Reviewer 2 Report

Journal: IJMS-1872960

Type of manuscript: Review

Title: Metabolic and cellular compartments of acetyl-CoA in the healthy and diseased brain

The authors systematically reviewed the fundamental role of mitochondrial acetyl-CoA derived from glycolytic glucose metabolism in the brain. Acetyl-CoA is regarded as the critical factor for neuronal and glial cells survival, not only because its central role in the TCA cycle and respiratory chain but also as a central factor for hundreds of acetylation reactions, including N-acetyl aspartate synthesis in neuronal mitochondria, acetylcholine synthesis in cholinergic neurons and different acetylation reactions for several proteins, peptides, histones and low molecular weight species in all cellular sub-compartments into the brain.

The review’s objectives and results are well presented and discussed, making the manuscript eligible for publication. However, there are some considerations the authors have to address.

1.- Line 46-47, the sentence: "On the other hand, glial cells, which constitute 80-90% of human brain cells." It could be interesting to point out that, at least in the neocortex, the complexity of astrocytes supporting the organization of neuronal circuitries occupy an estimated volume of around 20% of the total cell number in most adult brain regions [J. Neurosci. 2017, 37, 4493–4507.].

2.- Line 93: The inverse mechanism regulating GLUT1 concentrations on BBB capillaries by glucose concentration in the plasma. Would the authors include a brief sentence explaining how glucose levels in the blood up or down-regulate the concentrations of GLUT1 transporters? Are these mechanisms similar to those that regulate the concentration of GLUT3 transporter in neurons?

3.- Line 117: I do not understand the sentence. Is there a mistake in the following phrase? "However, in vivo lactate can (can not?) replace glucose entirely as energy precursor neither in physiological nor pathological conditions."

4.- Since pyruvate dehydrogenase complex (PDHC) is a mitochondrial enzyme redox-regulated by hydrogen peroxide and S-glutathionylation by the oxidation and reduction of specific sensitive cysteine residues [Redox Biol. 2013, 2, 123–139; Antioxidants. 2022; 11: 416], and acetyl-CoA is a direct precursor-substrate for the TCA cycle coupled with energy generation in the respiratory chain in the brain by PDHC from pyruvate originating from glycolytic metabolism of glucose or lactate oxidation by lactate dehydrogenase 1, could the author include a brief paragraph commenting this point and the importance of the redox regulation in PDHC and other sensitive cysteine redox enzymes in health and diseases in the brain?

For example, an important enzymatic complex involved in Alzheimer's disease pathophysiology is glyceraldehyde-3-phosphate dehydrogenase (GAPDH). Cytoplasmic GAPDH is a tetramer composed of four identical monomers containing a single sensitive-cysteine residue critical to the enzyme's catalytic function. This enzyme catalyzes the reversible phosphorylation of glyceraldehyde-3-phosphate involving the thiol group of Cys152 [J. Alzheimer's Dis. 2010, 20, 369–393]. The four sensitive cysteine residues of GAPDH can be oxidized by hydrogen peroxide, reducing the stability of the protein and resulting in monomers, dimers, and other denatured products. Interestingly, under pathological conditions, reduced glutathione can react with Cys152 contributing to the formation of disulfide bridges. Since GAPDH has been found oxidized in AD brains, it has been suggested that S-glutathionylation of the enzyme is a mechanism to protect the protein against irreversible damage in the oxidizing environment of the AD brain [J. Alzheimer's Dis. 2010, 20, 369–393; Antioxidants. 2022; 11: 416]. Could something similar happen with PDHC?

5.- Paragraph 243-265: Interestingly, data in this paragraph support the notion that exogenous acetate or its donors may be used as a complementary precursor of acetyl-CoA, bypassing the deficits of the PDHC step in different settings. Do the authors think that modulators of the redox proteome, like n-acetyl-cysteine, could contribute as an exogenous acetyl donor that could be used as a complementary precursor of acetyl-CoA?

6.- Subheading 3. "Origin and metabolic role of axonal acetyl-CoA," Because of the importance the axonal transport for survival, integrity, and proper neuronal function, it would be interesting to include a new figure to clarify the critical role of acetyl-CoA in the axonal anterograde and retrograde transport.

7.- Figures (1 and 2) attempt to clarify some of the concepts explained in the main text, but they are too complex and challenging to follow, mainly because of the number of arrows and symbols used on them. It could be more beneficial to separate some of these figures into two parts [(A) and (B)] in order to gain clarity.

Author Response

Referee 2

  1. We have described this finding and included complementary reference.(lines 53-56)
  2. We described mechanism of GLUT1 –glucose level interactions. (lines 106-109).
  3. Sentence was corrected. (lines 129-130)
  4. The problem of neuroprotectory activities of –SH groups of cysteine or glutathione has been included as short paragraph along with citations. (lines 574-584)
  5. The paragraph is now between lines 256-278. The role of acetate is described in greater details in chapter 6. We found no indications that acetyl-cysteine  may contribute to acetyl-CoA pool. On the other hand,  thiol compounds increase acetyl-CoA level (lipoamide, lipoic acid) by protecting PDHC and other enzymes with –SH groups in their active centres. (Ronowska et al.JNC 2007, 103, 972-983)
  6. Fig.  3 has been inserted with appropriate legend.
  7. The part of Fig. 1 content has been transferred into Fig. 4 demonstrating acetyl-CoA in endoplasmic reticulum and nucleus. Fig. 1 is now somewhat simpler. On the other hand, I would be happy for your kind acceptance Fig. 2, which presents summary of our findings on acetyl-CoA metabolism in cholinergic neurons under excitotoxic conditions.

I hope that our correction have met all points raised by reviewers.

Round 2

Reviewer 1 Report

N/A